# Long-lived topological time-crystalline order on a quantum processor

Liang Xiang [1,13], Wenjie Jiang [2,13], Zehang Bao [1,13], Zixuan Song [1], Shibo Xu [1], Ke Wang [1], Jiachen Chen [1], Feitong Jin [1], Xuhao Zhu [1], Zitian Zhu [1], Fanhao Shen [1], Ning Wang [1], Chuanyu Zhang [1], Yaozu Wu [1], Yiren Zou [1], Jiarun Zhong [1], Zhengyi Cui[1], Aosai Zhang [1], Ziqi Tan [1], Tingting Li [1], Yu Gao [1], Jinfeng Deng [1], Xu Zhang [1], Hang Dong [1], Pengfei Zhang [1], Si Jiang [2], Weikang Li [2], Zhide Lu [2], Zheng-Zhi Sun [2], Hekang Li [1], Zhen Wang [1,3], Chao Song [1], Qiujiang Guo [1,3] ✉, Fangli Liu [4,5], Zhe-Xuan Gong [6,7], Alexey V. Gorshkov [4], Norman Y. Yao [8], Thomas Iadecola [9,10], Francisco Machado [8,11], H. Wang [1,3] ✉ & Dong-Ling Deng [2,3,12] ✉

Topologically ordered phases of matter elude Landau's symmetry-breaking theory, featuring a variety of intriguing properties such as long-range entanglement and intrinsic robustness against local perturbations. Their extension to periodically driven systems gives rise to exotic new phenomena that are forbidden in thermal equilibrium. Here, we report the observation of signatures of such a phenomenon—a prethermal topologically ordered time crystal—with programmable superconducting qubits arranged on a square lattice. By periodically driving the superconducting qubits with a surface code Hamiltonian, we observe discrete time-translation symmetry breaking dynamics that is only manifested in the subharmonic temporal response of nonlocal logical operators. We further connect the observed dynamics to the underlying topological order by measuring a nonzero topological entanglement entropy and studying its subsequent dynamics. Our results demonstrate the potential to explore exotic topologically ordered nonequilibrium phases of matter with noisy intermediate-scale quantum processors.

Phases of matter are often classified by broken symmetries and local order parameters[1]. However, the discovery of topological order has transformed this simple paradigm[2,3]. Two topologically ordered phases with the same symmetries can showcase topologically distinct features, such as different patterns of long-range entanglement and the emergence of quasiparticles with different anyonic braiding statistics[4–6]. These features are intrinsically nonlocal in that they cannot be distinguished by any local order parameter[7,8] and also distinguish

[1]School of Physics, ZJU-Hangzhou Global Scientific and Technological Innovation Center, and Zhejiang Key Laboratory of Micro-nano Quantum Chips and Quantum Control, Hangzhou, China. [2]Center for Quantum Information, IIIS, Tsinghua University, Beijing 100084, China. [3]Hefei National Laboratory, Hefei 230088, China. [4]Joint Quantum Institute and Joint Center for Quantum Information and Computer Science, NIST and University of Maryland, College Park, MD, USA. [5]QuEra Computing Inc., Boston, MA, USA. [6]Department of Physics, Colorado School of Mines, Golden, CO, USA. [7]National Institute of standards and Technology, Boulder, CO, USA. [8]Department of Physics, Harvard University, Cambridge 02138 MA, USA. [9]Department of Physics and Astronomy, Iowa State University, Ames, IA 50011, USA. [10]Ames National Laboratory, Ames, IA 50011, USA. [11]ITAMP, Harvard-Smithsonian Center for Astrophysics, Cambridge Massachusetts 02138, USA. [12]Shanghai Qi Zhi Institute, Shanghai 200232, China. [13]These authors contributed equally: Liang Xiang, Wenjie Jiang, Zehang Bao. ✉e-mail: qguo@zju.edu.cn; hhwang@zju.edu.cn; dldeng@tsinghua.edu.cn

topologically ordered phases from symmetry protected topological phases[9]. Unfortunately, topological order is usually restricted to the ground state; mobile thermal excitations can hybridize nominally degenerate ground states by traversing the system along nontrivial closed loops. By introducing disorder, the motion of these excitations can be arrested and the hybridization process suppressed. In the limit where excitations are fully localized, the topological phase becomes stable across the entire energy spectrum of the system[10-18].

Time-periodic driving of a quantum many-body system enables novel phases of matter that cannot exist in thermal equilibrium. A prominent example is that of time crystals[19-26], where discrete time translation symmetry is spontaneously broken. Strikingly, the concept of a time crystal can be extended to include topological order, resulting in a new dynamical phase dubbed a topologically ordered time crystal[27]. Unlike conventional time crystals, where the breaking of time translation symmetry manifests in the dynamics of local observables, topologically ordered time crystals show such symmetry breaking only for nonlocal logical operators. Whether or not this phase has a truly infinite lifetime depends on the late-time stability of many-body localization[28-30]; nevertheless, the dynamical features of the system can still exhibit very long-lived signatures of localization persisting beyond current experimental timescales. While signatures of conventional time crystals without topological order have been observed in a number of distinct systems, including trapped ions[31,32], spins in nitrogen-vacancy centers[33,34], ultracold atoms[35,36], solid-state spin ensembles[37-39], and superconducting qubits[23,40,41], the observation of a topologically ordered time crystal remains an open challenge.

Here, we report the observation of a long-lived prethermal topologically ordered discrete time crystal, with eighteen programmable superconducting transmon qubits arranged on a two-dimensional square lattice. By optimizing the device fabrication and control process, we push the median lifetime of these qubits to $T_1 \approx 163\mu s$ and the median simultaneous single- and two-qubit gate fidelities above 99.9% and 99.4%, respectively. Together with a neuroevolution algorithm[42] that outputs near-optimal quantum circuits for digitally simulating four-body interactions, this enables us to successfully implement Floquet surface code dynamics with an optimized quantum circuit of depth exceeding 700, consisting of more than 2300 single- and 1400 two-qubit gates. We measure the dynamics of nonlocal logical operators and local spin magnetizations and find that the former show a robust subharmonic response, whereas the latter decay quickly to zero and do not show period-doubled oscillations. This differs drastically from symmetry breaking and symmetry-protected topological discrete time crystals, where local, rather than nonlocal, observables exhibit subharmonic response. We further reveal the long-range quantum entangled nature of topological order by preparing a many-body eigenstate of the Floquet unitary and measuring its topological entanglement entropy with different subsystem sizes and geometries[43,44]. We obtain near-expected values for the measured topological entanglement entropy, which deviates significantly from the trivial-state value of zero and provides strong evidence for the presence of topological order.

## Results

### Theoretical model and experimental setup

We consider the periodically driven rotated surface code model on a 2D lattice with open boundary conditions[27,45] (see Methods):

$$H(t) = \begin{cases} H_1, & 0 \le t < T', \\ H_2, & T' \le t < T, \end{cases}$$
$$H_1 \equiv \frac{\pi}{2}\sum_k \sigma_k^x + \sum_k \boldsymbol{B}_k \cdot \boldsymbol{\sigma}_k, \tag{1}$$
$$H_2 \equiv -\sum_p \alpha_p A_p - \sum_q \beta_q B_q.$$

where $\boldsymbol{\sigma}_k = (\sigma_k^x, \sigma_k^y, \sigma_k^z)$ is a vector of Pauli matrices acting on the $k$-th qubit; $\boldsymbol{B}_k$ denotes an on-site field drawn randomly and independently from a ball with radius $B$; the plaquette operators $A_p = \prod_{m \in p}\sigma_m^z$ and $B_q = \prod_{n \in q}\sigma_n^x$ are products of Pauli operators on the corresponding plaquettes (Fig. 1a); $\alpha_p$ and $\beta_q$ are coefficients uniformly chosen from $[0, 2\pi)$; the drive period is fixed as $T = 2T' = 2$, which roughly corresponds to a 1.4-$\mu$s runtime for the corresponding quantum circuit in our experiment.

We note that, other than the discrete time-translation symmetry, $H(t)$ breaks all microscopic symmetries due to the presence of the random on-site fields $\boldsymbol{B}_k$ in $H_1$. The Floquet unitary that fully characterizes the dynamics of the system reads $U_F = U_2 U_1$, with $U_1 = e^{-iH_1}$ and $U_2 = e^{-iH_2}$ being the unitary operators generated by the Hamiltonians $H_1$ and $H_2$, respectively. $H_2$ represents the Hamiltonian of the rotated surface code model, whose energy spectrum is two-fold degenerate and whose eigenstates show topological order[7,46]. Owing to their topological nature, the degenerate eigenstates can only be distinguished by nonlocal string operators such as $Z_L = \prod_{k \in P_z}\sigma_k^z$ or $X_L = \prod_{k \in P_x}\sigma_k^x$, which traverse the lattice through the path $P_z$ or $P_x$ (see Fig. 1a). We label each eigenstate pair by $|Z_L^{(l)} = \pm 1\rangle$ for each eigenstate with quasi-energy $\epsilon_l$ (see Supplementary Note 1.A). In the limit $B \to 0$, $U_1$ represents a perfect flip of all spins. As a result, the drive $H_1$ reorganizes the topologically ordered eigenstate pairs of $H_2$ into Floquet eigenstates $|E_\pm^{(l)}\rangle$ of the form $|E_\pm^{(l)}\rangle \propto |Z_L^{(l)} = 1\rangle \pm |Z_L^{(l)} = -1\rangle$. The quasi-energies of the corresponding cat-like eigenstates are split by quasi-energy $\pi$ (Fig. 1b). As a result, the stroboscopic dynamics of the nonlocal operator $Z_L$ exhibits a stable subharmonic oscillation with $2T$ periodicity as illustrated in Fig. 1c, which breaks the discrete time-translation symmetry by the drive period $T$ (see Supplementary Note 1.C). These Floquet eigenstates also exhibit topological order, which is essential for the robustness of the subharmonic response of the nonlocal string operators $Z_L$.

For small but finite $B$, the system's integrability is broken and the eigenstate pairs are no longer exactly split by $\pi$. However, this deviation arises from the motion of excitations across the system which mixes the different topological sectors, which is strongly suppressed by the disorder in $\alpha_p$ and $\beta_p$. Until this thermalization occurs, $t \lesssim t_{th}$, the system's dynamics will exhibit robust period doubling dynamics, much like in the $B = 0$ case. All our experimental and theoretical observations of period doubling behavior pertain to this "prethermal" regime which, in the small-$B$ regime, is much larger than the experimentally accessible timescales.

Our experiments are carried out on a programmable flip-chip superconducting processor with 18 transmon qubits arranged on a 2D square lattice (see Supplementary Note 2.A for detailed information about the device). To implement $H(t)$, the four-body terms with random strengths in $H_2$, which are vital for the eigenstate topological order at high energy, pose an apparent challenge since four-body interactions do not naturally appear in the superconducting system. We therefore exploit the idea of digital quantum simulation to implement $H(t)$ with quantum circuits (Fig. 1d), which are obtained via a neuroevolution algorithm[42] (see Methods and Supplementary Note 1.G). We mention that these quantum circuits are near-optimal and can implement $H(t)$ in an analytical fashion without any Trotter error, independent of $\alpha_p$, $\beta_p$, and $\boldsymbol{B}_k$. With these efficient quantum circuits, improved gate fidelities, and coherence times, we are able to implement and probe the unconventional dynamics of the system up to 20 driving periods.

### Subharmonic response for nonlocal observables

The characteristic signature of topological time-crystalline eigenstate order is the breaking of the discrete time-translation symmetry for nonlocal logical operators, manifested by persistent oscillations with period $2T$. To this end, we define the normalized auto-correlation function $A_L^{1/d}(t) = \text{sign}[\langle Z_L(0)Z_L(t)\rangle]|\langle Z_L(t)\rangle|^{1/d}$ for the $d$-body string

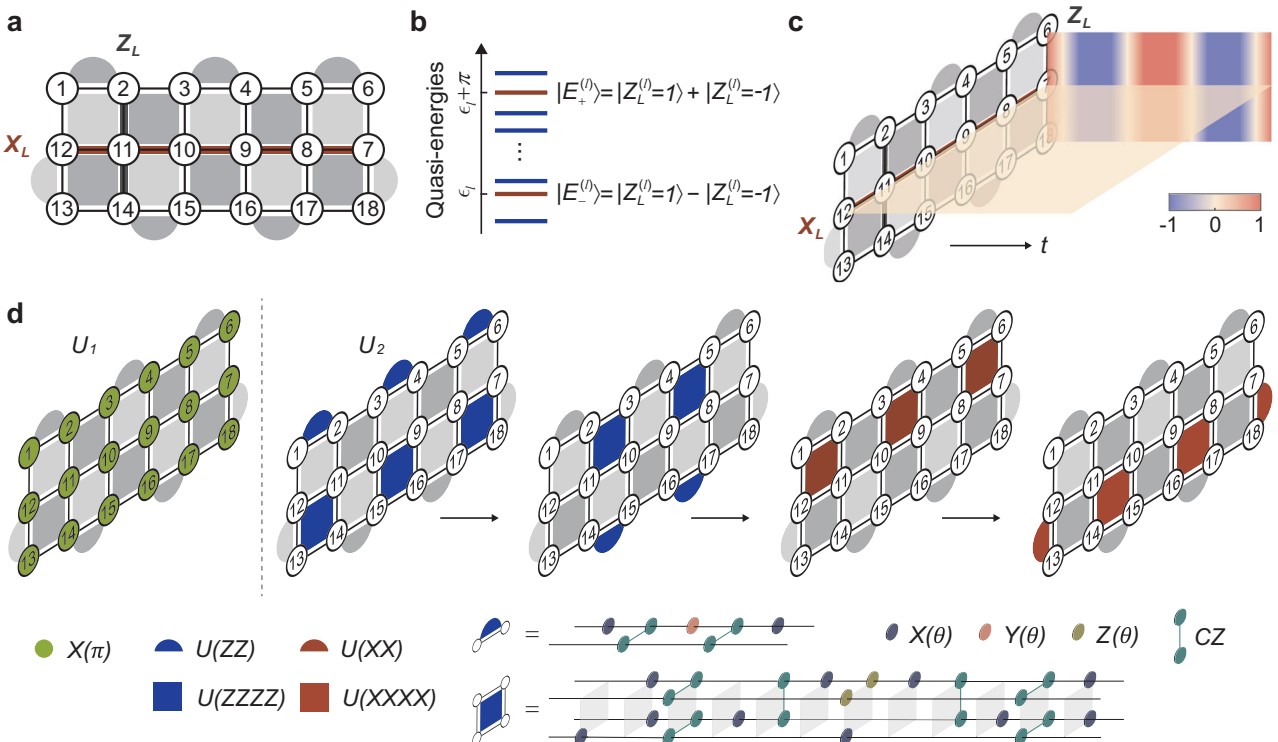

**Fig. 1 | Periodically driven surface code model. a** Rotated surface code model on a three-by-six square lattice. The circled numbers label the qubits. The dark and light gray regions represent plaquette operators $A_p$ and $B_q$, respectively. The thick black (red) line represents the nonlocal string operator $Z_L$ ($X_L$). **b** Topologically ordered Floquet eigenstates in the limit $B \to 0$. The quasi-energies of each pair of eigenstates $|E_\pm^{(l)}\rangle$ are split by $\pi$. **c** Schematic of the stroboscopic dynamics of the string operators $Z_L$ and $X_L$. Under periodic driving, the expectation value of $Z_L$ exhibits a persistent subharmonic oscillation with a period of $2T$, while $X_L$ preserves a constant value of zero. **d** Decomposition of the Floquet unitary $U_F$ ($B = 0$) into elementary

quantum gates. $U_1$ is realized by applying $\pi$ pulses to all the qubits. Since all the plaquette operators commute with each other, $U_2$ is constructed by sequentially applying four groups of them. Plaquette unitaries $e^{-iA_p T/2}$, labeled by $U(ZZZZ)$ and $U(ZZ)$, and $e^{-iB_q T/2}$, labeled by $U(XXXX)$ and $U(XX)$, are further decomposed into sequences of single-qubit rotations and two-qubit controlled-Z gates. $X(\theta)$, $Y(\theta)$, and $Z(\theta)$ denote single-qubit rotations by an angle $\theta$ around the x-, y-, and z-axis, respectively. $e^{-iB_q T/2}$ can be implemented by sandwiching $e^{-iA_q T/2}$ with Hadamard gates. In the experiment, the whole circuit is further compiled to reduce the depth and suppress hardware noise (Supplementary Note 2.E).

operator $Z_L$, where $\langle \cdots \rangle$ represents the expectation value and the $d$-th root is used to indicate the geometric mean value. When $d = 1$, this generalization reduces to the standard auto-correlation function for single-qubit operators[23,40], while allowing for a direct comparison across different lengths of string operators (see Supplementary Note 2.F for more details). We begin by studying the evolution of the disorder-averaged auto-correlator $\overline{A_{L_i}^{1/d}}(t)$ for operators $\{Z_{L_i}\}$ ($d = 3$) at the solvable limit $B = 0$, which are averaged over 24 random realizations by sampling Hamiltonian parameters $\alpha_p$, $\beta_q$ and random z-basis initial states. From Fig. 2a, it is evident that, in the topologically ordered regime, $\overline{A_{L_i}^{1/d}}(t)$ oscillates with a $2T$ periodicity for up to 20 driving cycles. We mention that $\overline{A_{L_i}^{1/d}}(t)$ exhibits a gradually decaying envelope due to extrinsic experimental imperfections, rather than internal thermalization, which is confirmed by numerical simulations (lines in Fig. 2a) incorporating experimentally measured gate errors and decoherence times. Indeed, the ideal numerical simulations show that the internal thermalization time of the system without experimental noise is far longer than 20 driving cycles (Supplementary Note 1.D). In the frequency domain, $\overline{A_{L_i}^{1/d}}$ shows a peak at the subharmonic frequency of the drive period $\omega/\omega_0 = 0.5$, as shown in Fig. 2b. We also note that the string operator $X_L$ does not show period-doubled oscillations, and no subharmonic peak is observed in the frequency domain.

Although the $2T$-period subharmonic oscillations of nonlocal observables $\{Z_{L_i}\}$ already sharply distinguish our experiment from

previous works[23,31–33,40], where only local observables break time-translation symmetry, we further demonstrate that the observed Floquet topological order is a nonlocal effect by contrasting with the dynamical behavior of local operators $\{\sigma_k^z\}$. The auto-correlation function $\overline{\langle \text{sign}[\sigma_k^z(0)]\sigma_k^z(t)\rangle}$ decays to zero quickly without evident oscillations (Fig. 2c), even though the periodic drive is locally applied to each qubit. The striking contrast between nonlocal operators $\{Z_{L_i}\}$ and local operators $\{\sigma_k^z\}$ exposes the locally indistinguishable nature of the Floquet topological order and rules out the possibility of trivial oscillations arising from driving a noninteracting system.

**Topologically ordered Floquet eigenstates**

The Floquet eigenstates bear intrinsic topological order and exhibit long-range quantum entanglement characterized by the topological entanglement entropy $S_{\text{topo}}$[43,44] (see Supplementary Note 1.B). To reveal the underlying global entanglement, we prepare an eigenstate of $U_F$ and measure its $S_{\text{topo}}$ for different system sizes. In the $B \to 0$ limit, eigenstates of $U_F$ correspond to superpositions of degenerate eigenstates of $H_2$ (see Fig. 1b). The eigenstate we prepare is the symmetric superposition of ground states of $H_2$, given by $|E_+^{(0)}\rangle = \frac{1}{\sqrt{2}}(|Z_L^{(0)} = 1\rangle + |Z_L^{(0)} = -1\rangle)$. We prepare it from a simple initial product state using a quantum circuit whose depth grows linearly with the system size (see Supplementary Note 1.H)[47]:

$$|E_+^{(0)}\rangle = \frac{1}{2^4}(1 + X_L)\prod_q(1 + B_q)|0\rangle^{\otimes 18}. \quad (2)$$

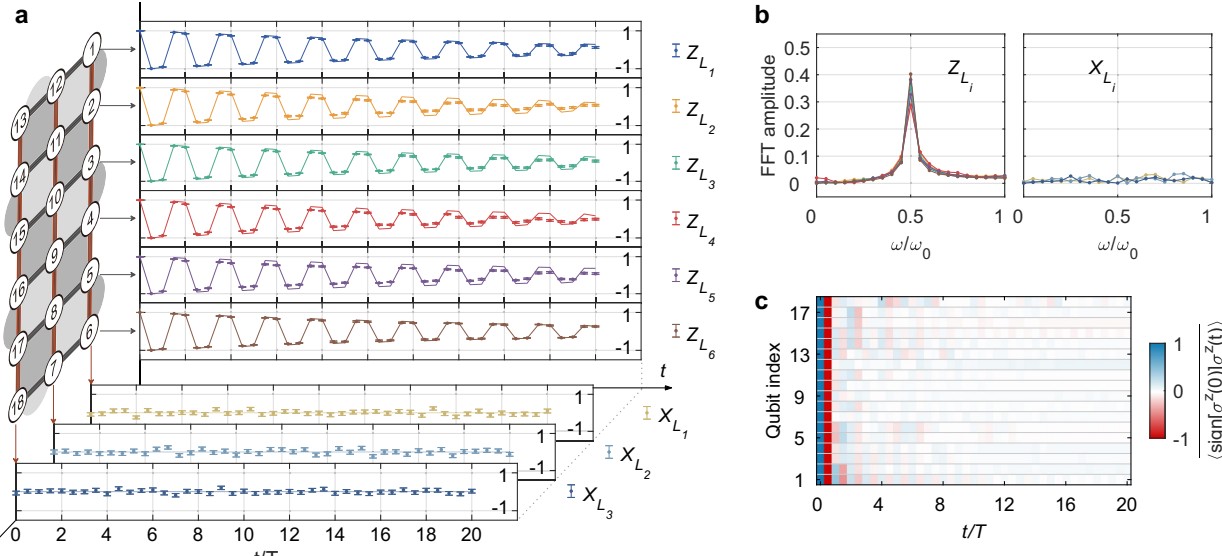

**Fig. 2 | Time-translation symmetry breaking for nonlocal observables with $B = 0$. a** Dynamics of nonlocal observables. Auto-correlation function for the three-body string operators $\{Z_{L_i}\}$ (thick black lines) and instantaneous expectation values for the six-body string operators $\{X_{L_i}\}$ (thick red lines) are shown in the upper six and lower three panels, respectively. Experimental data points (dots) are obtained from averaging over 24 random realizations, with error bars representing the standard error of the statistical mean. The numerical results (lines) are computed by taking into account qubit decoherence and gate errors (Supplementary Note 3). Whereas the instantaneous expectation values for $\{X_{L_i}\}$ remain zero, the auto-correlators for $\{Z_{L_i}\}$ exhibit stable subharmonic oscillations for up to 20 cycles (see Supplementary Note 2.F for measurement details). **b** Fourier spectra of time-domain signals observed in **a**, where a stable subharmonic frequency peak appears for $\{Z_{L_i}\}$ but not $\{X_{L_i}\}$. **c** Dynamics of the auto-correlation function for local observables $\{\sigma_k^z\}$. Such auto-correlations decay quickly to zero, in sharp contrast to those of the string operators $\{Z_{L_i}\}$.

We then measure the plaquette operators $\{A_p\}$ and $\{B_q\}$ (left panel of Fig. 3a), and an average value of ~ 0.95 is observed, which is noteworthy given that these operators encode four-body correlations. The high-fidelity gates and long coherence times achieved in our experiment are of crucial importance to obtain such a high average value of the measured stabilizers (see Supplementary Note 2.B). We also measure the expectation values of string operators $\{X_{L_i}, Z_{L_i}\}$ and find that $\langle E_+^{(0)}|Z_{L_i}|E_+^{(0)}\rangle \approx 0$ and $\langle E_+^{(0)}|X_{L_i}|E_+^{(0)}\rangle \approx 1$ (right panel of Fig. 3a). These experimental results are in good agreement with theoretical predictions, providing strong evidence that the prepared state is indeed a Floquet eigenstate as desired.

Having prepared the Floquet eigenstate, we further measure its topological entanglement entropy for two different subsystem sizes: four qubits and six qubits. We follow a protocol developed in ref. 47 and divide the subsystem into three parts: A, B, and C (upper panels of Fig. 3b). $S_{\text{topo}}$ can be extracted from the following combination of von Neumann entanglement entropies[43,44]:

$$S_{\text{topo}} = S_A + S_B + S_C - S_{AB} - S_{AC} - S_{BC} + S_{ABC}, \quad (3)$$

where $S_A$ is the von Neumann entropy for region $A$, while $AB$ means the union of regions $A$ and $B$, and similarly for other terms. For the eigenstates of $U_F$, the theoretically predicted value of $S_{\text{topo}}$ is $-\ln 2$[43]. For each region $i$, we perform quantum state tomography on the whole (four-qubit or six-qubit) subsystem and reconstruct $\rho_i$ to calculate the corresponding fidelity $F(\rho_i) = \text{tr}\sqrt{\sqrt{\rho_i}\rho_i^{\text{ideal}}\sqrt{\rho_i}}$ and von Neumann entropy $S_i = -\text{tr}(\rho_i \ln \rho_i)$, where $\rho_i^{\text{ideal}}$ is the reduced density matrix of region $i$ obtained by tracing out the complementary region of the ideal Floquet eigenstate. The experimentally measured $S_{\text{topo}}$, $S_i$, and $F(\rho_i)$ are shown in the lower panels of Fig. 3b. The measured von Neumann entropy for each region agrees well with the corresponding ideal value. In addition, we observe that $-S_{\text{topo}}/\ln 2$ is $0.86 \pm 0.02$ for the four-qubit and $0.84 \pm 0.08$ for the six-qubit subsystem, which is incompatible with the trivial-state value of zero and provides strong evidence for the nontrivial topological nature of

the prepared Floquet eigenstate. The deviation between the measured $S_{\text{topo}}$ and its corresponding ideal value is due to limited coherence times and gate errors, which is confirmed by numerical results using a noise model estimated via independent measurements (our numerical simulations show that $-S_{\text{topo}}/\ln 2$ is 0.85 and 0.82 for the four-qubit and six-qubit subsystems, respectively; see Methods and Supplementary Note 3).

**Robustness against local perturbations**

Topological order is expected to be robust against small local perturbations. In our experiment, we investigate the robustness of the subharmonic response of nonlocal logical operators and of the entanglement dynamics to local perturbations by turning on the random on-site fields in $H_1$. We vary the perturbation strength $B$ and measure $\overline{A_{L_i}^{1/d}}(t)$ and $S_{\text{topo}}(t)$, with results plotted in Fig. 4.

Figure 4a shows the measured disorder-averaged auto-correlation function $\overline{A_{L_i}^{1/d}}(t)$ for $\{Z_{L_i}\}$ under weak ($B = 0.1$) and strong ($B = 3.0$) perturbations, which are averaged over 24 realizations with randomly drawn initial states, $\alpha_p$, $\beta_q$, and $\boldsymbol{B}_k$. With a small perturbation ($B = 0.1$), $\overline{A_{L_i}^{1/d}}(t)$ continues to exhibit persistent subharmonic response up to 20 driving periods (upper panel of Fig. 4a), which is a defining feature of the time-translation symmetry breaking for nonlocal operators and shows the robustness of the observed prethermal topologically ordered discrete time crystal. In contrast, with a strong perturbation ($B = 3.0$), the measured $\overline{A_{L_i}^{1/d}}(t)$ decays quickly to zero and shows no subharmonic response (lower panel of Fig. 4a); at large $B$, the large onsite field rapidly destroys the topological order preventing any robust period doubling dynamics. To explore the crossover from the time-crystalline to trivial dynamics, we vary the perturbation strength $B$ and Fourier transform the measured time-domain signals. Fig. 4b shows the Fourier amplitudes at $\omega/\omega_0 = 0.5$ with $B$ ranging from 0 to 3.0. We find a small plateau at $B \lesssim 0.25$, which further supports the robustness of the topologically ordered time-crystalline dynamics

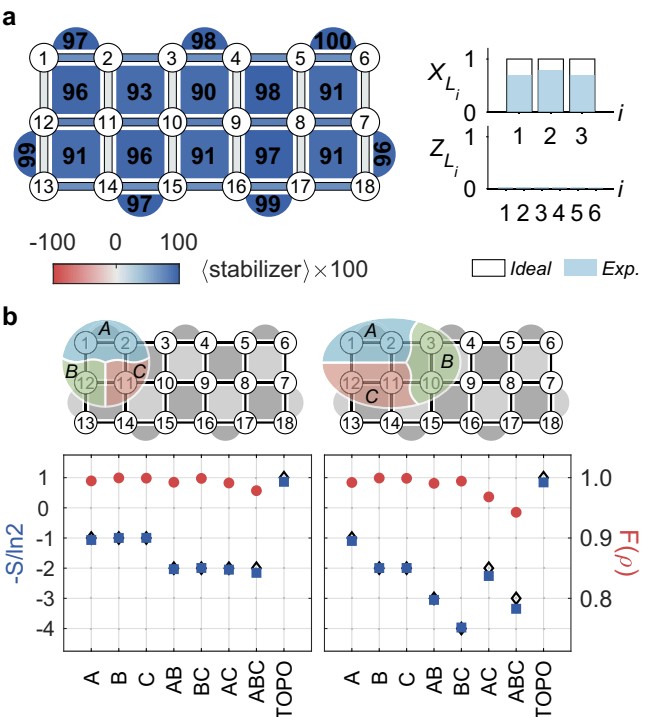

**Fig. 3 | Measured stabilizer values and topological entanglement entropy for a Floquet eigenstate. a** Measured expectation values of plaquette operators $\{A_p\}$ and $\{B_q\}$ for the Floquet eigenstate $|E_+^{(0)}\rangle$ are shown in the left panel. Expectation values of string operators $\{X_{L_i}, Z_{L_i}\}$ are plotted in the right panel with solid (hollow) bars representing experimental (theoretical) results. **b** Measuring topological entanglement entropy (TOPO, $S_{\text{topo}}$) on four- and six-qubit subsystems. The division of subsystems and corresponding experimental results are shown in the upper and lower panels, respectively. Blue squares (red circles) represent entanglement entropy (region fidelities $F(\rho_i)$). The topological entanglement entropy is extracted from the measured von Neumann entropy for the regions $S_i$. The error bars, obtained by repeated measurements, are very small and not shown here. The black rhombus markers are theoretical predictions for $S_{\text{topo}}$ and $S_i$, which agree well with the corresponding experimental results.

against weak perturbations. As $B$ increases, the Fourier amplitude decays monotonically and becomes almost flat at $B \gtrsim 2.5$, where the topological order is very quickly destroyed and no period doubling dynamics survives. Sample-to-sample amplitude fluctuations over 24 random realizations (inset of Fig. 4b) display a sharp increase at the same value of $B$ where the Fourier amplitude starts to decay, further highlighting the location of the crossover between the prethermal topological time-crystalline and the trivial dynamics.

We further study the dynamics of the topological entanglement entropy under local perturbations. We first prepare the system in the $B = 0$ Floquet eigenstate $|E_+^{(0)}\rangle$ and then let it evolve under $H(t)$ with varying $B$. In Fig. 4c, we plot the measured $\overline{S}_{\text{topo}}(t)$ for $B = 0.1$ and $B = 3.0$, respectively. From this figure, we see that $\overline{S}_{\text{topo}}(t)$ drops more quickly at strong perturbation $B = 3.0$ due to the breakdown of the topological phase. We note that $\overline{S}_{\text{topo}}(t)$ also has a slow decay even for $B = 0.1$ due to the accumulated gate errors in the circuit, which is confirmed by the numerical simulations (the dashed lines in Fig. 4c) (see Supplementary Note 3). In addition, we measure plaquette and string operators after evolution under $U_F$ for a single time step (see Fig. 4d). From this figure, it is clear that their values are largely preserved for $B = 0.1$, unlike the case of $B = 3.0$, where these values drop to near zero. We further measure the disorder-averaged $\overline{S}_{\text{topo}}(t = T)$ as a function of $B$ (Fig. 4e). Similar to the Fourier spectrum amplitudes in Fig. 4b, $\overline{S}_{\text{topo}}(t = T)$ also decays monotonically with increasing disorder

strength. Although, in general, the topological entanglement entropy tends to be destroyed by quench dynamics[48,49], the fact that it exhibits a plateau in the weak disorder regime ($B \lesssim 0.25$) implies a slow decay of the topological time-crystalline order, offering a different characterization of the topological time crystalline behavior and further validating its robustness against perturbations. We emphasize that the relationship between the observed nonzero topological entanglement entropy and the topological order in a general quantum state and away from equilibrium is complex. Here, we interpret the slow decay of $S_{\text{topo}}$ as an indicator of slow melting of the topological order. This is supported by theoretical arguments based on prethermalization and by experimental measurements of the fidelity between the topologically ordered initial state and the state obtained by evolution over up to five Floquet periods, which remains large provided $B$ is sufficiently weak (see Supplementary Note 1.D).

We note that, for the generic local perturbations considered in our experiment, the overlap between a bare logical operator and its corresponding dressed logical operator may vanish in the thermo-dynamic limit. This would render the observation of time-crystalline behavior for the bare logical operator infeasible[27]. In addition, to observe the time-crystalline behavior, it is also crucial that $\{\prod \sigma_k^x, Z_L\} = 0$ is satisfied, which requires that the length of $Z_L$ be odd. A possible way to maintain time-crystalline signatures in bare logical operators in the thermodynamic limit and to remove the requirement of odd length $Z_L$ is to consider a surface code with a hole, as discussed in depth in ref. 27. In our experiment, we do not adopt such a layout because measuring the corresponding nonlocal logical operator would become very challenging with the current device.

## Discussion

In summary, we have experimentally observed signatures of a long-lived topologically ordered time crystal in the prethermal regime with a programmable superconducting quantum processor. In contrast to previously reported conventional time crystals, the topologically ordered time crystal studied here builds upon a truly long-range entangled phase, such that discrete time-translation symmetry breaking only occurs for nonlocal logical operators. As a result, this experiment is more challenging than our previous one on symmetry-protected topo-logical (SPT) time crystals[23], requiring 2D qubit connectivity, deeper circuit depth, higher gate fidelity, and longer coherence time. With significantly improved quantum computing hardware (see Supplementary Table 2), we observed persistent sub-harmonic response for logical operators independent of the initial state and demonstrated the robustness of this response to generic perturbations without any microscopic symmetry. In addition, we also prepared a topologically ordered Floquet eigenstate and measured its topological entanglement entropy, which agrees well with theoretical predictions and clearly shows the intrinsic topological nature of the observed time crystal. Our work shows that the topologically ordered time crystal differs from previously studied time crystals[23,31-33,40] in terms of mea-surability, stability, and entanglement structure.

The topologically ordered eigenstates of the Floquet unitary are theoretically predicted to exhibit a perimeter law, where the expectation value of a Wilson loop scales with the perimeter rather than the area enclosed[17,27]. As a result, the late-time values of the nonlocal logical operators under the Floquet drive also exhibit a perimeter law, i.e., they will exponentially decay with their length (see Supplementary Note 1.E). However, it is still challenging to experimentally observe such long-time behaviors in current NISQ[50] devices, so we leave this for future experimental investigations. The high controllability and programmability of the superconducting processor demonstrated in our experiment also paves the way to exploring a wide range of other exotic non-

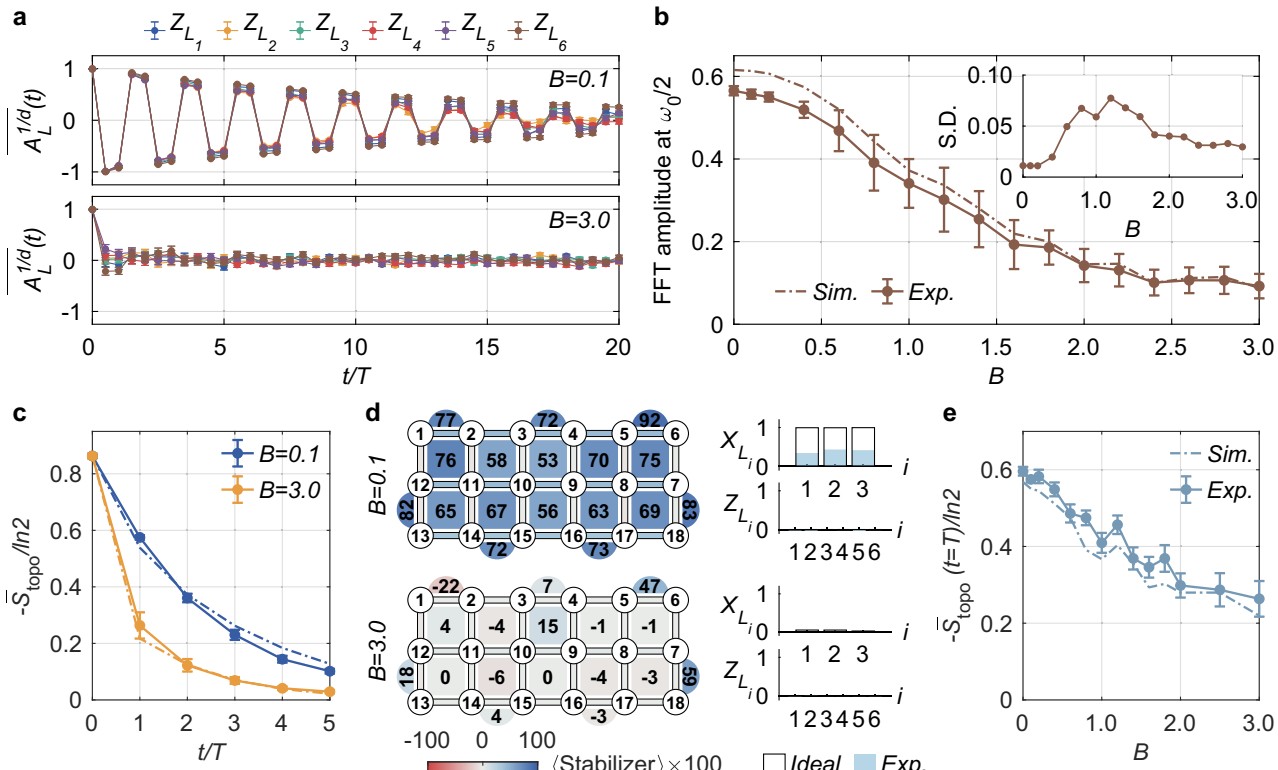

**Fig. 4 | Robustness of the topological time-crystalline eigenstate order.**
**a** Measured disorder-averaged auto-correlation function $\overline{A_L^{1/d}}(t)$ for string operators $\{Z_{L_i}\}$ with $B = 0.1$ (upper panel) and $B = 3.0$ (lower panel). Error bars denote the standard error of the statistical mean over 24 random realizations. **b** Amplitudes of Fourier spectra at $\omega/\omega_0 = 0.5$ as a function of $B$. Fourier transform of $A_L^{1/d}(t)$ is performed using averaged time-domain signals over $\{Z_{L_i}\}$ for up to $t = 6T$. Each data point is averaged over 24 random realizations. Error bars are the standard deviation (S.D.) for 24 disorder realizations. Insert: The S.D. of Fourier spectra amplitudes at $\omega/\omega_0 = 0.5$ as a function of $B$. **c** Quench dynamics of the disorder-averaged topological

entanglement entropy $\overline{S}_{\text{topo}}$ from the initial state $|E_+^{(0)}\rangle$ (which is a Floquet eigenstate at $B = 0$) for different $B$. Here, $\overline{S}_{\text{topo}}(t \neq 0)$ is obtained by performing state tomography on a four-qubit subsystem and the average is over 12 random realizations; $\overline{S}_{\text{topo}}(t = 0)$ is obtained via the same state tomography process and averaging over five repetitions of eigenstate preparation. **d** Measured plaquette and string operators for the eigenstate $|E_+^{(0)}\rangle$ after single-step $U_F$ evolution at $B = 0.1$ and $B = 3.0$. **e**, $\overline{S}_{\text{topo}}(t = T)$ as a function of random field strength $B$, which is averaged over 12 random realizations. Numerical simulations (dashed lines) in **b**, **c**, and **e** are carried out with noisy quantum gates (see Supplementary Note 3 for details).

equilibrium phases with intrinsic topological order that are not accessible in natural materials. In particular, it would be interesting and important to realize various dynamically-enriched topological orders[51]. Indeed, our experiment has demonstrated all necessary building blocks for implementing the Floquet-enriched topological order that hosts dynamical anyon permutation[51] and emergent non-Abelian anyons[52,53]. An observation of such an unconventional phenomenon would also mark an important step in deepening our understanding of exotic non-equilibrium phases.

## Methods
### Surface code model
Here, we provide a comprehensive introduction to the model, which we use to realize the long-lived topological time-crystalline order. Toric code is usually defined on a square lattice with periodic boundary condition. Thus, the defining manifold is a torus. Consider a $L \times L$ square lattice with periodic boundary condition. It contains $L^2$ vertices, $2L^2$ edges, and $L^2$ plaquettes. The qubit is put on each edge of the lattice and two kinds of local operators can be defined. For each plaquette, we can define the product of the $\sigma_k^z$ operators on each edge surrounding the plaquette as the corresponding plaquette operator. There are $L^2$ plaquette operators. For each vertex, we define the product of the $\sigma_k^x$ operators on each edge connecting with the vertex as the corresponding vertex operator. There are $L^2$ vertex operators. This definition gives a stabilizer code that can store quantum information.

However, in current quantum devices, qubits are usually connected with the nearest ones and periodic boundary condition is challenging to accomplish. Instead, stabilizer codes defined in open manifolds are more likely to be realized. Therefore, we concern with a square lattice with an open boundary condition.

To satisfy the requirements of a stabilizer code, the surface code on an open manifold has two kinds of boundaries in alternating order: the $z$-type boundary and the $x$-type boundary. At the $z$-type boundary, one of the qubits associated with the plaquette operators is removed, and at the $x$-type boundary, one of the qubits associated with the vertex operators is removed. Thus, in the bulk, the stabilizer operators are the same as those in the toric code. At the boundaries, the corresponding plaquette and vertex operators are only applied to the reserved three qubits.

We note that, for each pair of operators, if they are both the plaquette operators or vertex operators, they must commute with each other. Besides, if they belong to different classes, they must share an overlap with an even number of qubits (0 or 2). In addition with the anti-commutative relation $\{\sigma_k^z, \sigma_k^x\} = 0$, we conclude that any pair of operators from different classes commute with each other. Without modifying the topological properties, we can drop even more auxiliary qubits and rotate the surface code to obtain the model we used in the main text. For a $m \times n$ lattice, there are $mn$ qubits, $(m-1)(n-1)$ four-body stabilizers, and $m + n - 2$ two-body stabilizers. Thus, such code will encode a single qubit of quantum information (see Supplementary Note 1.A for details).

## Circuit

To implement the time evolution of surface code model, we decompose the evolution unitary into digital quantum circuits. It is straightforward to realize the Floquet drive $U_1(t) = e^{-itH_1}$ with tensor products of single-qubit rotations, which can be represented with Euler angles. However, the circuit construction of $U_2(t) = e^{-itH_2}$ is more challenging due to the two-body and four-body operators.

The variational quantum circuit is a powerful tool for NISQ computation and quantum simulation and has been intensively studied in recent years[54,55]. We adapt this method to construct the quantum circuit for the evolution. The circuit construction for the evolution operator of $H_2$ can be divided into two steps. First, we need to find an appropriate circuit ansatz with variational parameters. Second, we optimize the variational parameters in this ansatz to minimize the distance between the corresponding quantum circuit and the target unitary. In our work, we use the neuroevolution method[42] to find a suitable variational quantum circuit architecture. In short, we construct a directed graph where each node represents a block of quantum gates that can be implemented in parallel, and where the directed edges denote allowed sequences of blocks. A quantum circuit can then be represented as a directed path in this graph. We sample several paths from this graph, and use the gradient descent method to optimize their parameters. Using such a method, we find an experimentally friendly ansatz analytically representing the target evolution unitary (see Supplementary Note 1.G for details).

## Numerical simulation of the noisy circuit

We employ the Monte Carlo wavefunction method[56] to numerically simulate the noisy circuits. It requires fewer computational resources than the master-equation approach because it evolves the system state vector of size $2^N$ during the calculation rather than the density matrix of size $2^N \times 2^N$. In this context, we use the state-vector simulator provided by Qiskit[57] for the numerical calculation of system dynamics.

To model realistic errors in experimental circuits, we use quantum channels of energy relaxation, dephasing, and depolarizing. These channels are represented as probabilistic mixtures of different operators. Parameters of each error channel are estimated from the experimental benchmarks of gate errors and device performance (see Supplementary Note 3 for details). Error operators are sampled according to the noise model and randomly inserted after each ideal gate. As a result, the state vectors evolve along many quantum trajectories corresponding to many noise realizations. Values of the desired observable are obtained by averaging over an ensemble of such quantum trajectories, which resembles repeated measurements for evaluating the expectation value of an observable in experiments.

## Data availability

The data generated in this study have been deposited in the Zenodo database under accession code https://doi.org/10.5281/zenodo.13692134[58].

## Code availability

The simulation codes used in this study are available in the Code Ocean capsule at https://codeocean.com/capsule/8032749/tree/v1.

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

## Acknowledgements

We thank X.-J. Liu, L.-M. Duan, and D. Yuan for helpful discussions. The device was fabricated at the Micro-Nano Fabrication Center of Zhejiang University. We acknowledge the support from the Innovation Program for Quantum Science and Technology (Grant Nos. 2021ZD0300200 and 2021ZD0302203), the National Natural Science Foundation of China (Grant Nos. 92365301, 12274368, 12075128, T2225008), the Zhejiang Provincial Natural Science Foundation of China (Grant No. LR24A040002), and Zhejiang Pioneer (Jianbing) Project (No. 2023C01036). T.I. acknowledges support from the National Science Foundation under Grant No. DMR-2143635. F.M. acknowledges support from the NSF through a grant for ITAMP (Award No. 2116679) at Harvard University. A.V.G. was supported in part by the NSF QLCI program (award No. OMA-2120757). N.Y.Y. acknowledges support from the U.S. Department of Energy via the National Quantum Information Science Research Centers Quantum Systems Accelerator, from the Army Research Office (Grant NO. W911NF-24-1-0079) and from a Simons Investigator award. W.J., S.J., W.L., Z.L., Z.-Z.S., and D.-L.D acknowledge in addition support from the Tsinghua University Dushi Program and Shanghai Qi Zhi Institute.

## Author contributions

L.X. and Z.B. carried out the experiments and analyzed the experimental data under the supervision of Q.G. and H.W.; W.J., L.X., S.J., and Z.B. performed the numerical simulations under the supervision of D.-L.D., Q.G., F.L., F.M., Z.-X.G., A.V.G., and T.I.; D.-L.D., W.J., F.L., F.M., Z.-X.G., A.V.G., N.Y., T.I., W.L., Z.L., and Z.-Z.S. conducted the theoretical analysis; H.L. and J.C. fabricated the device supervised by H.W.; D.-L.D., Q.G., W.J., L.X., H.W., F.L., Z.-X.G., A.V.G., F.M., and T.I. co-wrote the manuscript; H.W., Q.G., C.S., Z.W., L.X., Z.B., Z.S., S.X., K.W., J.C., F.J., X.Z., Z.Z., F.S., N.W., C.Z., Y.W., Y.Z., J.Z., Z.C., A.Z., Z.T., T.L., Y.G., J.D., X.Z., H.D., and P.Z. contributed to experimental setup. All authors contributed to the discussions of the results.

## Competing interests

The authors declare no competing interests.
