## [Peer Review File · Nature Communications]

Long-lived topological time-crystalline order on a quantum processorREVIEWER COMMENTS

Reviewer #1 (Remarks to the Author):

In this manuscript, the authors realize topological time crystals on a superconducting quantum processor. The experiment is based on the Floquet drive of a two-dimensional spin model with the surface code Hamiltonian and random local magnetic fields. The discrete time-translation symmetry of the system is spontaneously broken during the pre-thermal time evolution. According to accompanying numerical simulations, the pre-thermal subharmonic oscillations are stabilized via many-body localization for much longer times than the experimental timescales. In contrast to standard Floquet time crystals, the subharmonic oscillations are observed for nonlocal observables corresponding to the logical operators of the surface code.

The manuscript is well written. Measured data are clearly presented and supported by numerical simulations. This work is scientifically sound and it contributes to the development of quantum computing technologies and the investigation of the prethermal behavior of many-body systems.

The experiment and accompanying numerical simulations demonstrate the stability and measurability of topological time crystals. This is an important result, which remained unanswered in the associated theoretical proposal [30]. On the other hand, topologically non-trivial time crystals, in particular featuring symmetry-protected topological order, have been theoretically discussed in several previous works (see a review [Harper et al., *Annu. Rev. Condens. Matter Phys.* 11, 345–368 (2020)]) and experimentally demonstrated in Ref [13] by the authors of this paper. The level of innovation compared to these existing works remains unclear. Novelty in terms of hardware capabilities is also unclear as the experiment [13] is implemented on the same type of superconducting chip using the same native gates. The clarification of the differences from previous works is thus necessary to fully assess this manuscript.

Questions concerning novelty and innovation:

1) The authors write in the introduction that “topologically ordered time crystals remain an open challenge” prior to this work. However, topologically non-trivial time crystals, in particular featuring symmetry-protected topological order, have been theoretically discussed in several previous works (see a review [Harper et al., *Annu. Rev. Condens. Matter Phys.* 11, 345–368 (2020)]) and experimentally demonstrated in Ref [13] by the authors of this paper. What are the key differences between SPT order and intrinsic topological order in time crystals? What are their implications for the measurability and stability of the time crystals? Answering these questions is crucial for assessing the novelty of this work.

In section “Subharmonic response for nonlocal observables”, the authors argue that “...subharmonic oscillations of non-local observables $\{Z_{Li}\}$ already sharply distinguish our experiment from previous works [13, 34–36, 43], where only local observables break time-translation symmetry”. However, SPT time crystals feature subharmonic oscillations in string order parameters that are also nonlocal [C. W. von Keyserlingk and S. L. Sondhi, *Phys. Rev. B* 93, 245145 (2016)]. Thus, the non-locality of order parameters does not distinguish topological order from SPT order.

2) Does this experiment exploit any hardware capabilities that go beyond the previous experiment [13] realized by the authors of this paper? The chip architecture is the same in this work as in Ref [13]: frequency-tunable transmons arranged in a two-dimensional 6x6 grid with nearest-neighbor connectivity facilitated by tunable couplers. Moreover, the experiment exploits the same native gates: single-qubit rotations and two-qubit CZ gates. It seems that this experiment simply uses a smaller number of qubits, 18 compared to 26 in Ref [13], that are available on the superconducting chip. Can the authors clarify the innovations of the quantum computing hardware?

Furthermore, we have several scientific and technical questions that should be addressed before the publication of this manuscript in any journal:

1) For time crystals realized on NISQ devices, it is important to distinguish the decay of correlation functions due to external decoherence (hardware imperfections) and due to the internal thermalization of the spin array. The characterization of external decoherence, for example via echo sequences consisting of the Floquet unitary and its inverse, is crucially missing. Such characterization allows for clearly distinguishing external decoherence from internal thermalization as demonstrated in Refs. [13] and [43], and it is thus a key technique for detecting time-crystals on NISQ devices.

2) The autocorrelation functions of non-local operators plotted in Fig. 2a and Fig. 4a are very non-standard. They are different from order parameters defined in the theoretical proposal [30] as well as from standard auto-correlation functions typically used for time crystals. In numerical simulations presented in the supplementary material, the authors study the dynamics of nonlocal operators themselves (instead of these autocorrelation functions). Moreover, it is unclear, how these nonstandard autocorrelation functions are measured in the experiment. Since the subharmonic oscillations of the non-local order parameters are the main signature of topological time crystals, the definition of these autocorrelations, their relation to standard autocorrelation functions and their measurement should be clarified.

3) As explained above, the distinction from previous studies of SPT time crystals is unclear. It would be thus appropriate to support this experiment with a further analysis of unique features of the topologically ordered time crystals. It has been predicted in the theoretical proposal [30] that the nonlocal order parameters of the topologically ordered time crystals inherit the “perimeter law” from their equilibrium counterparts. However, this parameter law has not been studied in this experiment. Based on the numerical simulations presented in the supplementary material, the spin array should remain in a prethermal regime for relevant experimental timescales already for 3x3 and 5x3 grids. One could thus study the Z_L operator for two different lengths 3 and 5 to probe the perimeter law and demonstrate unique features of topological order.

In summary, this manuscript is an interesting addition to the literature studying pre-thermal behavior of quantum many-body systems. However, the clarification of the differences from previous works on SPT time crystals is necessary to assess whether the manuscript meets the standards of Nature Communications in terms of novelty and innovation.

Reviewer #1 (Remarks on code availability):

The repository contains data for all plots as well as codes for numerical simulations allowing for the reproduction of the numerical results.

Reviewer #2 (Remarks to the Author):

Reviewer #2 (Remarks on code availability):

Reviewer #3 (Remarks to the Author):

Xiang et al. experimentally study a topologically ordered time crystal by means of a digital quantum simulation in an array of superconducting transmon qubits. They demonstrate period-doubled oscillations of the expectation values of a nonlocal logical operator, and measure the topological entanglement entropy of a prepared quasienergy eigenstate. They also investigate how these features behave when a perturbation is introduced to the periodic driving protocol.

This is a convincing demonstration of a previously-unobserved dynamical phase, and the circuit optimizations which have been made in order to reach large simulation depths are interesting. However, some of the analyses made and the conclusions drawn from them are insufficiently justified. The points below should be addressed.

1. It is not clear what the quantity $S_{\text{topo}}(t)$ ---the topological entanglement entropy after a quench in the evolution parameters---is showing. In clean systems, it does not seem that the topological entanglement entropy is always constant after a quench. See [Zeng et al. Phys. Rev. B 94, 125104 (2016)]. Even in an MBL system, the post-quench state will generically evolve to one with volume-law entanglement. Many-body localization only implies that the entanglement grows logarithmically with time. As such, the interpretation of $S_{\text{topo}}(t)$ in terms of showing stability of topological order seems unmotivated (and no references are given to justify this interpretation). Do the authors have any reason for studying $S_{\text{topo}}(t)$? What is the theoretical picture of its ideal behavior, assuming that topological time crystalline order persists both before and after the quench, but with post-quench Floquet eigenstates that have very small overlap with the pre-quench states?

2. The "nonlocal observables" shown to have period doubling in this work are three-body, which is quite local. What is the prospect of measuring subharmonic response in observables with larger support? For instance, it should be possible to use the same device but modify H1 (of equation 1) and measure a period-doubled oscillation in the six-body XL (measurements of which are already reported in Fig. 2a).

3. The Floquet eigenstate of equation 2 is prepared independently of the actual Floquet circuit. The authors are relying on the theoretical characterization of the circuit they are trying to implement in order to make any comparison between their target state and the Floquet evolution. The authors should demonstrate that their prepared state is indeed a Floquet eigenstate for the evolution they experimentally achieve by, for instance, showing that the expectation value of XL is (close to) a constant as a function of the number of Floquet cycles.

4. What initial state is used to compute the autocorrelation functions $A^{\{1/d\}}_L(t)$? This does not seem to be stated anywhere.

List of changes

(All changes are marked in red in the main text and the Supplementary Information.)

1. We have added discussions on the difference between the topologically ordered time crystal and the SPT time crystals in the main text.
2. We have added discussions on the topological entanglement entropy in the main text.
3. We have added discussions on the perimeter law in the main text.
4. We have added a new supplementary section (Supplementary Section I.E) to explain the perimeter law and its experimental challenges. A new figure (Figure S5) is included.
5. We have added a new supplementary section (Supplementary Section I.F) to compare the topologically ordered phases studied in this work with the symmetry-protected topological phases studied in the previous work Ref.[13]. A new table (Table S1) is included.
6. We have added a new supplementary section (Supplementary Section II.A) to highlight the improvements in quantum computing hardware compared with our previous study in Ref.[13].
7. We have added a new supplementary section (Supplementary Section II.F) to provide the details of the definition of the auto-correlation functions for nonlocal operators, their relation to that of a single qubit, and their measurement in the experiment.
8. We have modified Fig. S3 to agree with the definition of auto-correlation.
9. We have added other necessary revisions throughout the whole manuscript to improve the presentation and address the Referees' comments/suggestions.

Point-by-Point Response

Response to Reviewer # 1 (Remarks to the Author):

We sincerely thank the Reviewer for their careful reading of our manuscript and for their assessment of our work as “an important result,” that “contributes to the development of quantum computing technologies and the investigation of the prethermal behavior of many-body systems”. We also greatly appreciate their thoughtful suggestions, which we have incorporated in our manuscript, improving it significantly. Most notably, we have added more in-depth discussions and clarifications to emphasize and highlight that: (a) our work realizes a novel dynamical phase which is fundamentally distinct from previous work; (b) our experiment exploits the advanced hardware capabilities which arise from a significant improvement over the experiments reported in previous work Ref. [13]. We believe that given these improvements to our manuscript, it is now suitable for publishing in Nature Communications. The detailed response to the Reviewer’s comments (point by point) is provided below.

Comment 1.1 In this manuscript, the authors realize topological time crystals on a superconducting quantum processor. The experiment is based on the Floquet drive of a two-dimensional spin model with the surface code Hamiltonian and random local magnetic fields. The discrete time-translation symmetry of the system is spontaneously broken during the pre-thermal time evolution. According to accompanying numerical simulations, the pre-thermal subharmonic oscillations are stabilized via many-body localization for much longer times than the experimental timescales. In contrast to standard Floquet time crystals, the subharmonic oscillations are observed for nonlocal observables corresponding to the logical operators of the surface code.

The manuscript is well written. Measured data are clearly presented and supported by numerical simulations. This work is scientifically sound and it contributes to the development of quantum computing technologies and the investigation of the prethermal behavior of many-body systems.

The experiment and accompanying numerical simulations demonstrate the stability and measurability of topological time crystals. This is an important result, which remained unanswered in the associated theoretical proposal [30]. On the other hand, topologically non-trivial time crystals, in particular featuring symmetry-protected topological order, have been theoretically discussed in several previous works (see a review [Harper et al., Annu. Rev. Condens. Matter Phys. 11, 345–368 (2020)]) and experimentally demonstrated in Ref [13] by the authors of this paper. The level of innovation compared to these existing works remains unclear. Novelty in terms of hardware capabilities is also unclear as the experiment [13] is implemented on the same type of superconducting chip using the same native gates. The clarification of the differences from previous works is thus necessary to fully assess this manuscript.

Response 1.1 We thank the Reviewer for their time and their assessment that our manuscript is “well written” and “clearly presented and supported by numerical simulation”. In particular, the Reviewer also explicitly pointed out the significance of our work as “it contributes to the development of quantum computing technologies and the investigation of the prethermal behavior of many-body systems.”

The Reviewer’s main concerns arise from the distinction between this work and the previous literature (in particular Ref. [13]), as well as the apparent lack of experimental technical developments. We thank the Reviewer for raising these points, which help us sharpen the presentation of our manuscript and better frame our results. Although we cover each of these points in detail in Responses 1.2 and 1.3, respectively, here we already want to emphasize that, indeed, our current work and Ref. [13] greatly differ in terms of the observed phase of matter, as well as the underlying hardware capabilities. Whereas Ref. [13], considered the time crystalline behavior in a symmetry-protected topological phase, where topologically protected *local* edge modes exhibit a robust sub-harmonic response; in the present work we investigate the time crystalline behavior within *topologically*

ordered state, where the sub-harmonic response can only be observed in truly non-local string operators that cross the entire system. Despite both sharing the terminology *topological* they are very distinct phenomena, whose differences we highlight in Table 1. The nature of this more intricate phase, as well as the study of a more complex observable, required a significant improvement in the quality of the superconducting chip, which we highlight in Table 2.

Questions concerning novelty and innovation:

Comment 1.2 The authors write in the introduction that “topologically ordered time crystals remain an open challenge” prior to this work. However, topologically non-trivial time crystals, in particular featuring symmetry-protected topological order, have been theoretically discussed in several previous works (see a review [Harper et al., Annu. Rev. Condens. Matter Phys. 11, 345–368 (2020)]) and experimentally demonstrated in Ref [13] by the authors of this paper. What are the key differences between SPT order and intrinsic topological order in time crystals? What are their implications for the measurability and stability of the time crystals? Answering these questions is crucial for assessing the novelty of this work.

In section “Subharmonic response for nonlocal observables”, the authors argue that “. . . subharmonic oscillations of non-local observables Z_{Li} already sharply distinguish our experiment from previous works [13, 34–36, 43], where only local observables break time-translation symmetry”. However, SPT time crystals feature subharmonic oscillations in string order parameters that are also nonlocal [C. W. von Keyserlingk and S. L. Sondhi, Phys. Rev. B 93, 245145 (2016)]. Thus, the non-locality of order parameters does not distinguish topological order from SPT order.

Response 1.2 We thank the Reviewer for raising this point. While both symmetry-protected topological (SPT) order (the focus of Ref. [13]) and topological order share the nomenclature of “topology”, they are very distinct phenomena with different signatures and properties. This is a point that Reviewer #3’s recognizes when highlighting that our work “is a convincing demonstration of a previously-unobserved dynamical phase...”. Some of their distinguishing features can be summarized as follows:

1. Fundamentally, SPT states and topologically ordered (TO) states belong to different quantum phases [see X.-G. Wen, Rev. Mod. Phys. 89, 041004 (2017) and X. Chen *et al.*, Phys. Rev. B 82, 155138 (2010)] and exhibit fundamentally distinct properties. Naming a few that of relevance to our work:
 - *Stability*: SPT phases (while topological in the sense that there is no local observable that can distinguish them from the trivial state) require the presence of a symmetry to be defined; as a result, any symmetry breaking term immediately destroys the SPT phase. By contrast TO phases are robust to *any* perturbation and do not require any underlying symmetry to be well-defined, making them more robust and fundamental. A distinct perspective on this point comes in terms of the preparation of such states: while a TO ordered state can only be generated with a unitary circuit whose depth scales with the system size, for the SPT phase, this is only true if all the gates respect the underlying symmetry, and thus shorter preparation schemes exist if the symmetry is not respected during the preparation. For this reason TO is referred to having *long-range entanglement (LRE)*, whereas SPT only *short-range entanglement (SRE)*.
 - *Nature of ground state degeneracy*: While both phases exhibit a ground state manifold whose degeneracy depends on the topology of the system itself, their origin is very different. In the case of an SPT, the symmetry acts in a non-trivial way in the system such that it ensures the existence of localized gapless edge modes. For example, in the one-dimensional spin-1 AKLT system, this manifests itself as the existence of two decoupled spin-1/2 degrees of freedom in the edge. As such, different ground states can be accessed by acting with *local* operators on the edge. By contrast, in a TO phase, the degeneracy of the ground state is not just encoded in the edge degrees of freedom. One can see that in terms of the operators required to transform between the different ground

states: the simplest example (which is of relevance to our work as well) is the toric code where the necessary string operators span the entire system size.

Based on these characteristics, the SPT and TO phases are distinct and as such one can sketch a possible phase diagram for local gaped quantum systems, Fig. 1(a) [see also in Chen *et al.* Phys. Rev. B 82, 155138 (2010)]. Crucially, the Floquet eigenstates of the SPT time crystal in Ref. [13] belong to short-range entangled phases [SRE, box with blue boundary in Figure 1(a)], while the Floquet eigenstates of the TO time crystal in this work belong to long-range entangled phases [LRE, box with red boundary in Figure 1(a)].

2. SPT time crystal in Ref. [13] and TO time crystal in this work show subharmonic response in qualitatively different types of operators. As illustrated in Ref. [13] and [Kumar *et al.*, Phys. Rev. B 97, 224302 (2018)], while the *non-local* string order parameter can diagnose the presence of the SPT phase, the subharmonic response of time-crystalline behavior is only present in *local edge operators* [see Figure 1(b)]. Indeed, the oscillation period of the non-local string order parameter, i.e., $O_{sg} = \sigma_1^z \sigma_2^y \left(\prod_{k=3}^{k=N-2} \sigma_k^x \right) \sigma_{N-1}^y \sigma_N^z$, is the same as the Hamiltonian period, while the local edge spin exhibits the sub-harmonic response characteristic of the time crystalline phase.

In contrast, for the TO time crystal explored in this work, only non-local string operators manifest a subharmonic time-crystalline response (see Fig. 2 of the main text), with local edge operators quickly decaying (we further illustrate this point in Fig. 5 of the response when addressing the comments of Reviewer #3). Indeed, this is the most prominent difference between the TO time crystal and previously studied SPT time crystals.

3. SPT time crystal and TO time crystal exhibit different robustness to perturbations as they inherit the robustness of the underlying type of topological phase. More specifically, the SPT time crystal is robust to small local *symmetric* perturbations [see Harper *et al.*, Annu. Rev. Condens. Matter Phys. 11, 345–368 (2020)], because such a phase is only protected when the symmetry is present. However, the TO time crystal is robust to arbitrary small local perturbations. This ensures that the TO time crystalline behavior is much less fine-tuned and thus more robust.
4. The Floquet eigenstates for SPT time crystal and TO time crystal have different entanglement structure. Topological entanglement entropy quantifies the long-range entanglement. Floquet eigenstates for SPT time crystal are SPT states, which are short-range entangled [SRE in Figure 1(a)] and have zero topological entanglement entropy. By contrast, Floquet eigenstates of TO time crystal are TO states, which host anyons and are long-range entangled [LRE in Figure 1(a)]. The long-range entanglement was experimentally measured by a non-zero topological entanglement entropy (see FIG. 3 of the main text).
5. As mentioned by the Reviewer, the “perimeter law” of the nonlocal order parameter is a unique feature of the TO time crystal, which has no counterpart in SPT time crystals. Wahl *et al.* (arXiv:2105.09694) states that the late-time values of the nonlocal logical operator Z_L exponentially decay to the length of the corresponding string. We numerically simulate the quenched dynamics of the TO time crystal for different system sizes, and show that the late-time envelopes of the nonlocal operators Z_L agree well with the perimeter law (see Figure 3).

In summary, the TO time crystal studied in this work is a novel dynamical phase which was not observed in previous literature. Such a phase is different from the SPT time crystals, in the measurability, stability, and entanglement structure (see Table 1). To experimentally probe its dynamics and exotic nonlocal topological features, both theoretical and experimental improvements were accomplished in this work, which justify the novelty and innovation of our manuscript.

We have added additional discussions of this distinction to our manuscript, and included a new section in supplementary information (Supplementary Section I.F) to provide a detailed comparison.

Figure 1: (a) Phase diagram for local gaped quantum systems (modified from FIG.3 of the paper [Chen et al. Phys. Rev. B 82, 155138 (2010)]). Our work focuses on the TO time crystal, which belongs to the long-range entanglement (LRE) one (box with red boundary). (b) Time evolution of the string order parameter and the local edge spin for an SPT time crystal in Ref. [13]. For simplicity, we focus on the drive protocol with $\delta = V_k = h_k = 0$, and initialize the system as one of the eigenstates for O_{sg} , which is also a product state with finite local edge spin.

	Phases	Measurability	Perimeter law	Stability	Ent. structure
TO time crystal	TO	Non-local operator	Yes	Arbitray perturbation	LRE
SPT time crystal	SPT	Local operator	No	Symmetric perturbation	SRE

Table 1: Comparison between the topologically ordered time crystal and Floquet symmetry-protected topological phases. Here, Entanglement structure is abbreviated as Ent. structure, long-range entanglement is abbreviated as LRE, and short-range entanglement is abbreviated as SRE.

Comment 1.3 Does this experiment exploit any hardware capabilities that go beyond the previous experiment [13] realized by the authors of this paper? The chip architecture is the same in this work as in Ref [13]: frequency-tunable transmons arranged in a two-dimensional 6x6 grid with nearest-neighbor connectivity facilitated by tunable couplers. Moreover, the experiment exploits the same native gates: single-qubit rotations and two-qubit CZ gates. It seems that this experiment simply uses a smaller number of qubits, 18 compared to 26 in Ref [13], that are available on the superconducting chip. Can the authors clarify the innovations of the quantum computing hardware?

Response 1.3 We thank the referee for this question. In short, yes; the performance of the quantum computing hardware has been significantly improved in this work, which plays a vital role in enabling this challenging experiment. Simulating TO time crystal with digital quantum circuits requires ~ 700 layers of quantum gates on a 2D array, which is about 3 times longer than that in Ref. [13] (~ 240 layers in a 1D chain). Therefore, it is necessary to improve gate fidelity, coherence time, and control electronics so that we can successfully observe the true TO time crystals before the accumulated circuit errors dominate the system dynamics and totally destroy the quantum coherent behavior.

We have summarized the improvements of quantum computing hardware in Table. 2. Compared with Ref. [13], in this work, the energy relaxation time T_1 is ~ 5 times longer, two-qubit gate errors are ~ 3 times lower, single-qubit gate errors are ~ 10 times lower, the sequence length is ~ 3 times longer, and the DAC capabilities are also largely improved. Putting all these improvements together has been a challenging experimental achievement that enables the study of the TO time crystal.

Following the Reviewer’s suggestion, we explicitly clarify these improvements in the revised supplementary information.

Hardware parameters	Ref. [13] [Nature 607, 468 (2022)]	This work
T_1 (median)	33.0 μs	162.6 μs
Maximum circuit length	10.4 μs	28.8 μs
Two-qubit gate connectivity	1D chain	2D lattice
Single-qubit gate error	0.55×10^{-2}	0.48×10^{-3}
CZ gate error	1.57×10^{-2}	0.64×10^{-2}
Number of CZ layers	160	420
Number of SQ layers	80	300
DAC board waveform storage	$\sim 15 \mu\text{s}$	$\sim 130 \mu\text{s}$
DAC resolution	14 bits	16 bits
DAC sampling rate	1×10^9 Hz	2×10^9 Hz

Table 2: Hardware improvements exploited in this study compared with the previous work Ref. [13].

Furthermore, we have several scientific and technical questions that should be addressed before the publication of this manuscript in any journal:

Comment 1.4 For time crystals realized on NISQ devices, it is important to distinguish the decay of correlation functions due to external decoherence (hardware imperfections) and due to the internal thermalization of the spin array. The characterization of external decoherence, for example via echo sequences consisting of the Floquet unitary and its inverse, is crucially missing. Such characterization allows for clearly distinguishing external decoherence from internal thermalization as demonstrated in Refs. [13] and [43], and it is thus a key technique for detecting time-crystals on NISQ devices.

Response 1.4 We thank the Reviewer for pointing out this technical question. The echo method is indeed an important technique to distinguish the external decoherence from the internal thermalization. However, its implementation requires a doubling of the length of the experimental sequence, resulting in 1440 layers of quantum gates and a total sequence time of 57.6 μs . This requirement is beyond the reach of our current NISQ device.

Another general method to evaluate the effects of external decoherence is to measure gate errors and decoherence times with standard benchmarks, such as cross-entropy benchmarking (XEB), energy relaxation time T_1 , and spin-echo dephasing time T_2^{SE} , and then perform noisy numerical simulations with these error sources. In our work, we carefully benchmarked these errors (Supplementary Section II.B) and used an open source framework (IBM Qiskit) to perform the numerical simulations (Supplementary Section III). The numerical simulation of the experiment is in excellent agreement with the experimental data (Fig. 2 of the main text), indicating a thorough understanding of the decoherence mechanisms in our experiments.

We complement this analysis by numerically simulating the echoed sequence [$U_{\text{echo}} = (U_F^\dagger)'(U_F)'$] mentioned by the Reviewer. Figure 2 displays the numerical results of the Z_{L_i} operators dynamics (grey lines), the auto-correlation functions of which are defined as $A_L^{1/3(\text{echo})}(t) = \text{sign}[\langle Z_L(0)Z_L(t) \rangle] |\langle Z_L(t) \rangle|^{1/(3 \times 2)}$ (note that an additional square root is taken to account for the twice longer sequence of U_{echo}). We find that $A^{1/3(\text{echo})}$ accurately captures the observed decay in both our previous numerical simulation as well as experimentally measured data, providing a strong indication that the observed decay is dominated by external sources of decoherence rather than internal thermalization. These results, included in the supplementary, further emphasize the robustness of the TO time crystalline behavior observed.

Figure 2: Numerical results of the echoed dynamics for $\{Z_{L_i}\}$ under $U_{\text{echo}} = (U_F^\dagger)^l (U_F)^l$. The auto-correlation functions $A^{1/3(\text{echo})}$ for U_{echo} (grey lines) accurately capture the observed decay of $A^{1/3}$ in both our previous numerical simulation (colored lines) as well as experimentally measured data (dots). Values of $A^{1/3(\text{echo})}$ are averaged over the same 24 random realizations as that under U_F . The plotted auto-correlation functions $A^{1/3}$ for U_F replicate the data points in Fig. 2a of the main text.

Comment 1.5 The auto-correlation functions of non-local operators plotted in Fig. 2a and Fig. 4a are very non-standard. They are different from order parameters defined in the theoretical proposal [30] as well as from standard auto-correlation functions typically used for time crystals. In numerical simulations presented in the supplementary material, the authors study the dynamics of nonlocal operators themselves (instead of these auto-correlation functions). Moreover, it is unclear, how these nonstandard auto-correlation functions are measured in the experiment. Since the subharmonic oscillations of the non-local order parameters are the main signature of topological time crystals, the definition of these auto-correlations, their relation to standard auto-correlation functions and their measurement should be clarified.

Response 1.5 We thank the Reviewer for raising this concern. While we agree that our definition of the auto-correlation function is not standard, as we will argue now, it is the natural choice for comparing the auto-correlation function across different string operators. To unpack this idea, let us start by re-introducing the auto-correlation function:

$$A_L^{1/d}(t) = \text{sign}[\langle Z_L(0)Z_L(t) \rangle] |\langle Z_L(t) \rangle|^{1/d} \quad (1)$$

where Z_L is the string operator of length d . The motivation for this quantity is then two-fold: First, as emphasized in a different response, the string operator of a topologically ordered phase is expected to approach a steady state whose values decay exponentially with the *length* of the operator. As a result, if one directly compares $(-1)^l \langle Z_L(0)Z_L(t) \rangle$ for different lengths of strings, they will decay to very different values, *even though the underlying physical phenomena is the same*. Using a “geometric mean” weighting by taking d -th root of the value of the correlator normalizes this length decay and thus allows us to directly compare the dynamics

of string operators of different sizes. Of course, this operation becomes ill-defined if the value is negative, so we separate the sign information from the amplitude information by multiplying, a posteriori, by the sign of the initial value. Second, one important feature of this quantity is that it is compatible with the single-spin auto-correlation function that is usually defined when studying other forms of time crystalline behavior: $\Phi(t) = \langle \sigma(0)\sigma(t) \rangle$ [see Else *et al.* *Annu. Rev. Condens. Matter Phys.* 11:467–99 (2020)].

Having motivated how we have defined the auto-correlation function, we now turn to its measurement in the experiment. In our work, we focus on the nonlocal operator $M = Z_L$ which exhibits time-crystalline behavior. Since the initial state is a product state along z -basis, $|\langle Z_L(0)Z_L(t) \rangle| = |\langle Z_L(t) \rangle|$ and the auto-correlation function can be rewritten as $\text{sign}[\langle Z_L(0)Z_L(t) \rangle] |\langle Z_L(t) \rangle|^{1/d}$, as we describe in the main text. For the non-local operator $M = X_L$, which is orthogonal to the initial state, and thus, if we do not consider the experimental noise, $\langle X_L(0)X_L(t) \rangle = 0$. To be consistent with the geometric mean, we define $\text{sign}[\langle X_L(t) \rangle] |\langle X_L(t) \rangle|^{1/d}$ as the generalized auto-correlation of X_L .

We experimentally measured the dynamics of $Z_L(t)$ and $X_L(t)$ under the Floquet dynamics for up to 20 cycles. The expectation value of $Z_L(t)$, given by $\text{tr}(\rho(t) \prod_{k \in P_z} \sigma_k^z)$, is determined by the diagonal elements of the density matrix $\rho(t)$, which can be obtained by simultaneously measuring all relevant qubits in the z -axis. $X_L(t)$ is measured by applying a $-\pi/2$ rotation around the y -axis to each qubit (mapping the x -axis to the z -axis) before the final measurements along the z -axis. To ensure the statistical accuracy for the measurements of $Z_L(t)$ and $X_L(t)$, we repeat the state initialization, evolution sequence, and measurements 10,000 times, making the sample size large enough for accurately estimating probability distributions of 3 or 6 qubits. Their auto-correlation functions are then averaged over 24 realizations of parameters α_p, β_q , and initial product states.

Comment 1.6 As explained above, the distinction from previous studies of SPT time crystals is unclear. It would be thus appropriate to support this experiment with a further analysis of unique features of the topologically ordered time crystals. It has been predicted in the theoretical proposal [30] that the nonlocal order parameters of the topologically ordered time crystals inherit the “perimeter law” from their equilibrium counterparts. However, this parameter law has not been studied in this experiment. Based on the numerical simulations presented in the supplementary material, the spin array should remain in a pre-thermal regime for relevant experimental timescales already for 3x3 and 5x3 grids. One could thus study the Z_L operator for two different lengths 3 and 5 to probe the perimeter law and demonstrate unique features of topological order.

Response 1.6 We agree with the Reviewer that the “perimeter law” is one of the unique features of the topological order. Its experimental observation will also help us to distinguish the topological order, in addition to the demonstration of the subharmonic response for non-local logical operators and the non-zero topological entanglement entropy. However, this extra experimental demonstration is very challenging beyond the capability of our current NISQ devices.

Wahl *et al.* (arXiv:2105.09694) states that the expectation value of Wilson loop operator after long-time evolution will exponentially decay regarding to its perimeter [see also in Bauer *et al.* *J. Stat. Mech.* P09005 (2013)]. In our work, we adopted the surface code model with boundaries, and thus the measured non-trivial logical operator is not closed. The perimeter law indicates that its late-time value will exponentially decay to the length of the corresponding string. Although the hardware parameters of our devices have been largely improved from those in previous experiments, we stress that experimental observation of the perimeter law is still very challenging in current NISQ devices for the following reasons:

1. The timescale required to experimentally measure the late-time values of those nonlocal operators still largely exceeds the maximum timescale that the current devices can simulate.
2. For our experiments on the current NISQ device, the decay of the logical operators is mainly caused by experimental errors, including gate errors, decoherence errors, etc, but not the spread of the local information. This will significantly affect the measured scaling behavior for the non-local operators.

3. Another way to probe the perimeter law is to measure the expectation values of the nonlocal logical operators for the Floquet eigenstates. However, it requires experimentally preparing the eigenstates of Floquet unitary with perturbations. Such states are locally modified by the details of those perturbations, and finding analytical quantum circuits to prepare remains an important open challenge.

Even if we are able to overcome these challenges, from the measurement of the nonlocal operators Z_L for only two different lengths $||Z_L|| = 3, 5$ we would be unable to claim the observation of a perimeter law. Unfortunately, this is simply too little data to confidently determine the nature of the decay and distinguish an exponential decay (i.e., the perimeter law) from any other scaling. To obtain more data points, simulating larger systems is required, which is very challenging due to the limited qubit coherence time and gate fidelity of current quantum devices.

To this end, we numerically simulate the quenched dynamics of the TO time crystal for different system sizes $N = 2l$, where l is the length of the measured non-local operator Z_L . We record the dynamics of non-local logical operators (see Figure 3a), and show their absolute values at critical time $t^* = 100T$ to probe the perimeter law (see Figure 3b). We find an evident exponential scaling for Z_L with different lengths, as predicted by the perimeter law.

We thank the Reviewer for raising this important point. To address this concern, we added a more in-depth discussion about the perimeter law in the main text, and also added a new section in supplementary information (Supplementary Section I.E) to include these numerical results and their associated description.

Figure 3: Decay of the logical string operator Z_L . **a**, Time evolution for Z_L for different lengths. **b**, Exponential scaling of Z_L at t^* .

Comment 1.7 In summary, this manuscript is an interesting addition to the literature studying pre-thermal behavior of quantum many-body systems. However, the clarification of the differences from previous works on SPT time crystals is necessary to assess whether the manuscript meets the standards of Nature Communications in terms of novelty and innovation.

Response 1.7 In summary, we greatly appreciate the Reviewer's helpful suggestions/comments, which have guided us to improve the manuscript significantly, especially for the clarification of the novelty and innovation. We have substantially revised the manuscript to clarify the differences between the previous work and the improvement of experimental hardware. We believe that this work is important for the rapidly growing fields of quantum computing, quantum simulation, and non-equilibrium phases of matter. We hope the strengthened

manuscript will satisfy the Reviewer and convince them to recommend its publication in Nature Communications.

Reviewer #1 (Remarks on code availability):

The repository contains data for all plots as well as codes for numerical simulations allowing for the reproduction of the numerical results.

Thank you for checking. We fully support the ongoing efforts to promote an open science environment.

Reviewer #2 (Remarks to the Author):

I co-reviewed this manuscript with one of the Reviewers who provided the listed reports. This is part of the Nature Communications initiative to facilitate training in peer review and to provide appropriate recognition for Early Career Researchers who co-review manuscripts.

Response We sincerely thank the Reviewer for their careful reading of our manuscript. In the revised manuscript, we have carefully and thoroughly addressed all the comments/suggestions from the Reviewer and improved the manuscript accordingly. We believe it is now suitable for publishing in Nature Communications.

Reviewer #2 (Remarks on code availability):

I co-reviewed this manuscript with one of the Reviewers who provided the listed reports. This is part of the Nature Communications initiative to facilitate training in peer review and to provide appropriate recognition for Early Career Researchers who co-review manuscripts.

Response We sincerely thank the Reviewer for his/her check.

Response to Reviewer #3 (Remarks to the Author):

We thank the Reviewer for judging our experimental results “a convincing demonstration of a previously-unobserved dynamical phase” and for pointing out that “the circuit optimizations which have been made in order to reach large simulation depths are interesting”. We also appreciate their valuable suggestions which have helped us improve the paper. We take these comments and suggestions very seriously and have substantially revised our manuscript to answer the following points: (a) how the measurement of the topological entanglement entropy after a quench helps us understand the stability of the topologically ordered time crystalline order; (b) how the prepared Floquet eigenstate is validated. The newly added discussions improve the paper significantly and we believe it is now suitable for publishing in Nature Communications. The detailed response to the Reviewer’s comments is provided below.

Comment 3.1 Xiang et al. experimentally study a topologically ordered time crystal by means of a digital quantum simulation in an array of superconducting transmon qubits. They demonstrate period-doubled oscillations of the expectation values of a nonlocal logical operator, and measure the topological entanglement entropy of a prepared quasienergy eigenstate. They also investigate how these features behave when a perturbation is introduced to the periodic driving protocol.

This is a convincing demonstration of a previously-unobserved dynamical phase, and the circuit optimizations which have been made in order to reach large simulation depths are interesting. However, some of the analyses made and the conclusions drawn from them are insufficiently justified. The points below should be addressed.

Response 3.1 We thank the Reviewer for the accurate summary of the main results of our manuscript. Indeed, to our knowledge, this is the first experimental demonstration of topologically ordered time crystals. We are

very excited about the completion of this experiment and believe it will have a far-reaching impact on future investigation for complex non-equilibrium phases on NISQ devices.

Comment 3.2 It is not clear what the quantity $S_{\text{topo}}(t)$ —the topological entanglement entropy after a quench in the evolution parameters—is showing. In clean systems, it does not seem that the topological entanglement entropy is always constant after a quench. See [Zeng et al. Phys. Rev. B 94, 125104 (2016)]. Even in an MBL system, the post-quench state will generically evolve to one with volume-law entanglement. Many-body localization only implies that the entanglement grows logarithmically with time. As such, the interpretation of $S_{\text{topo}}(t)$ in terms of showing stability of topological order seems unmotivated (and no references are given to justify this interpretation). Do the authors have any reason for studying $S_{\text{topo}}(t)$? What is the theoretical picture of its ideal behavior, assuming that topological time crystalline order persists both before and after the quench, but with post-quench Floquet eigenstates that have very small overlap with the pre-quench states?

Response 3.2 We thank the Reviewer for raising this important question. We agree with the Reviewer that the quenched dynamics will generically evolve the state to one with volume-law entanglement, even for an MBL system. However, focusing on small times, this quantity remains useful in estimating the breakdown of the Floquet topological order. When the perturbation strength is increased, the localization length will also increase until the topological order melts; at this point, the measured topological entanglement entropy quickly approaches zero.

Precisely, we regard the topological entanglement entropy as a nonlocal order parameter and investigate this crossover as a probe of the robustness of the topological order to local perturbations. When we initialize the system in a topologically ordered Floquet eigenstate of the unperturbed system ($B = 0$), the post-quench state will remain unchanged when $B = 0$. For small B and strong disorder in α_p, β_q , the topological entanglement entropy of the post-quench state will decay slowly and an overall logarithmic growth of the entanglement entropy is expected (as stated by the Reviewer). By contrast, for large B , the topological entanglement entropy of the post-quench state quickly decays, and the subharmonic response of non-local logical operators will also quickly melt. This serves as an independent, observable way of probing the topological time crystal (i.e. without knowing a priori the form of the correct string operators that exhibit the sub-harmonic response).

In FIG. 4c and e of the main text, it is clear that the decay rate monotonically increases with the growth of the perturbation strength B and shows a plateau at $B \lesssim 0.25$, which agrees well with the above theoretical analysis. To clarify this point, we have highlighted the physical picture in the revised main text and mentioned related references to avoid any possible misunderstanding in the scope of our observations.

Comment 3.3 The "nonlocal observables" shown to have period doubling in this work are three-body, which is quite local. What is the prospect of measuring subharmonic response in observables with larger support? For instance, it should be possible to use the same device but modify H1 (of equation 1) and measure a period-doubled oscillation in the six-body XL (measurements of which are already reported in Fig. 2a).

Response 3.3 While we agree with the referee that measuring the subharmonic response in longer string operators would be even more appealing, we believe that the subharmonic response of the three-body operators exhibited in our study is sufficient to identify the non-local characteristic of the topologically ordered time crystal. Indeed, the $2T$ -period subharmonic oscillation is only observed if we consider the full three-body Z_L string operators. To emphasize this point, we now contrast the long-lived sub-harmonic response of the string operators Z_L to both one σ_k^z and two-body local $\{\prod_{k \in P_{x_1}, k \notin P_{x_2}} \sigma_k^z\}$ operators, related to the original string Z_L operators. In both cases, we observe a very fast decay to zero with no subsequent dynamics; in Figure 4, we depict the auto-correlation function of the two-body operator case, denoted as $\overline{A_{L_i}^{1/2}}(t)$.

With regards to considering bigger system sizes, it is important to note that a larger string operator showing the DTC behavior in the surface code model must have a support of at least five qubits. The larger support of

the observables of interest necessarily implies a larger susceptibility to the qubit coherence time and the gate fidelity. To illustrate this point, we conduct noisy numerical simulations of the Floquet surface code dynamics on a 5×3 qubit lattice, wherein Plaquette operators of A_p and B_q interchange to manifest a period-doubled oscillation in the five-body Z_L operator (Note that the six-body X_L operator will not oscillate even when we modify H_1 , since its support contains an even number of qubits). The noise applied to this new circuit is the same as the simulation adopted to the 3×6 lattice. Figure 5 shows that the subharmonic oscillations of the five-body Z_L operators decay more rapidly compared to the three-body ones, limiting the oscillation cycles of auto-correlator to ~ 10 periods. Given this enhanced decay, we opt to focus on the three-body string operator which already demonstrates the non-local nature of the sub-harmonic response while not being as affected by external decoherence as these longer string operators. In the future, we hope to explore these larger-scale dynamics in more detail.

Figure 4: Measured dynamics of the auto-correlation function for two-body operators $\{\prod_{k \in P_z, k \notin P_{x_2}} \sigma_k^z\}$. The data points are obtained by averaging over 24 random experimental realizations. Error bars represent the standard error of the statistical mean.

Comment 3.4 The Floquet eigenstate of equation 2 is prepared independently of the actual Floquet circuit. The authors are relying on the theoretical characterization of the circuit they are trying to implement in order to make any comparison between their target state and the Floquet evolution. The authors should demonstrate that their prepared state is indeed a Floquet eigenstate for the evolution they experimentally achieve by, for instance, showing that the expectation value of XL is (close to) a constant as a function of the number of Floquet cycles.

Response 3.4 We thank the Reviewer for raising this important concern. We agree with the Reviewer that the Floquet eigenstate of Eq. 2 is prepared independently of the evolution circuit, and should be carefully considered. To show it is a Floquet eigenstate for $B = 0$, we measured the plaquette operators and the nonlocal string operators. We found all plaquette operators $\{A_p, B_q\}$ are nearly equal to one, logical operators $\{Z_{L_i}\}$ are

Figure 5: Numerical simulated dynamics of 5-body operators in a 5×3 qubit lattice ($\{Z_{L_i}\}$). The auto-correlation function is defined as $(\text{sign}[\langle Z_{L_i}(0)Z_{L_i}(t) \rangle] |\langle Z_{L_i}(t) \rangle|^{1/5})$. The same noise model and parameters are used in this simulation as those to simulate the 3×6 lattice. The results are averaged over 24 random realizations.

nearly equal to zero, and logical operator $\{X_{L_i}\}$ are nearly equal to one (see Fig. 3a of the main text). These results agree well with the theoretical expectation and validate that it is indeed a Floquet eigenstate. A similar method is also adopted in [Satzinger *et al.* Science 374, 1237 (2021)].

Dynamics of the plaquette operators and nonlocal string operators were measured in Fig. 4d of the main text, where we showed that the plaquette operators and logical operators $\{X_{L_i}\}$ have significant values after one period of Floquet drive under small perturbations ($B = 0.1$). The decay of those values is mainly caused by the inevitably experimental noise, without which, the plaquette operators $\{A_p, B_q\}$ and non-local string operators $\{X_{L_i}, Z_{L_i}\}$ are nearly constant after time evolution. This is another evidence that the prepared state is indeed a Floquet eigenstate. We stress that due to the accumulated gate errors in the experiment, the values for those many-body operators will decay quickly to zero if we evolve the system for more periods. Thus, showing that those values are close to constants as a function of the number of Floquet drives is very challenging in NISQ devices, and we left it for future investigation.

Comment 3.5 What initial state is used to compute the auto-correlation functions $A_L^{1/d}(t)$? This does not seem to be stated anywhere.

Response 3.5 We thank the Reviewer for raising this concern. We randomly chose product states polarized along z -basis as initial states. We now explicitly mention it in the revised main text.

In summary, we greatly appreciate the Reviewer's careful reading of the manuscript. Based on their reports, we have added a substantial amount of contents to improve the presentation significantly. We have carefully addressed all the comments/suggestions raised by the Reviewer. We hope this significantly enhanced manuscript will satisfy the Reviewer and convince them to recommend the publication of this work in Nature Communications.

REVIEWER COMMENTS

Reviewer #1 (Remarks to the Author):

We would like to thank the authors for their exhaustive response to our comments. They have addressed all our comments by the revision of the manuscript and the supplementary material as well as by a detailed point-by-point response.

They clarified that experimental improvements compared to the previous work [13] included significantly longer coherence times, larger gate fidelities and improved control electronics. This was crucial for implementing deep quantum circuits required for realizing topological time crystals. They also clarified differences between the topologically ordered time crystals realized in this work and the Floquet symmetry-protected topological time crystals from their previous paper. These differences include the subharmonic response of global observables, robustness against all kinds of local perturbations and long-range entanglement. They motivated and explained the autocorrelation function of the logical Z operator. Its measurement in the experiment has also been clarified. The authors have convincingly answered our questions about the perimeter law and echo sequences by numerical simulations presented in the supplementary material.

We conclude that this work notably contributes to the investigation of the prethermal behavior of many-body systems. It presents impressive experimental results demonstrating the capabilities of the superconducting quantum processor. As all technical points have been convincingly addressed, we recommend the manuscript for publication in Nature Communications, provided that the authors address one remaining issue explained below that emerged in the revised manuscript.

Technical comment:

In the revised supplementary material, the authors clarify that the autocorrelation function $\langle X_L(0)X_L(t) \rangle$ of the logical X_L operator is not measured in the experiment. Instead, only the instantaneous value of X_L is measured. This is not problematic in itself, as they

observe the subharmonic response of the autocorrelation function $\langle Z_L(0)Z_L(t) \rangle$ of the logical Z_L operator. However, in the caption of Fig. 2 and on page 15 of the supplementary material, the authors interpret the instantaneous value of X_L as its auto-correlation function. This interpretation is misleading and should be corrected. Moreover, on page 15 of the supplementary material, the authors claim that $\langle X_L(0)X_L(t) \rangle = 0$. This is not generally true. For vanishing on-site field $B = 0$, the logical X_L operator is a conserved quantity as it commutes with both H_1 and H_2 . As it squares to unity, $\langle X_L(0)X_L(t) \rangle = 1$ for any initial state in contrast to $\langle X_L(t) \rangle = 0$ for an initial computational basis state. This discrepancy indicates that interpreting the instantaneous value of X_L as its autocorrelation function is incorrect.

Reviewer #2 (Remarks to the Author):

Reviewer #3 (Remarks to the Author):

I am satisfied with most of the responses of the authors to the previous round of reports. I did not realize that Fig. 4d already showed the expectation value of X_L after one Floquet step, which addresses my concern about the relation of the Floquet eigenstate to the actual evolution as realized by the experiment.

However, I am still concerned about the interpretation of $S_{\text{topo}}(t)$. As I understand it, the authors' response and addition to the main text appeals to the notion that $S_{\text{topo}}(t)$ being close to the surface code value of $\ln(2)$ means that the state is close to a surface code state (or one with the same topological order). Thus, the slow decay of $S_{\text{topo}}(t)$ could be interpreted as diagnosing the slow melting of topological order.

This interpretation of the topological entanglement entropy is an assumption, and should be stated as such. At the very least, the authors have skipped many steps in justifying this interpretation. If the state were guaranteed to be area law entangled and pure, then this interpretation would likely be valid. But a mixed and potentially volume law entangled state seems like it should be able to have any value of $S_{\text{topo}}(t)$ without necessarily having any relation to the corresponding topological order. $S_{\text{topo}}(t)$ in this paper goes through the value $-0.5 \ln(3)$, but the state is certainly not related to the $\nu=1/3$ Laughlin state.

That $S_{\text{topo}}(t)$ decays faster for $B=3.0$ than $B=0.1$ is a completely generic feature of making a stronger perturbation, and the same reasoning could have been applied to any observable which starts at 1 for $B=0$ and evolves to 0 after a quench to $B>0$. It does not imply that the numerical value of $S_{\text{topo}}(t)$ quantifies the extent to which the state remains topologically ordered.

If the authors insist that the topological entanglement entropy does quantify the remnant topological order in their state, then they must argue for this more carefully, and at least include their analysis in the supplemental material. Otherwise, it should be clearly stated that the relationship between the slow decay of $S_{\text{topo}}(t)$ and the existence of perturbatively dressed logical operators is an assumption which they do not intend to address in this work.

List of changes

(All changes are marked in red in the main text and the Supplementary Information.)

1. We have modified the caption of Fig. 2 in the main text to clarify that the values plotted here are the instantaneous values of the X_L operators.
2. We have corrected the misleading interpretation of X_L operators and provided measurement details for the X_L operators plotted in Fig.2 of the main text.
3. We have added discussions about the relationship between topological entanglement entropy and the topological order in the main text.
4. We have provided a theoretical analysis of the state evolution under the Floquet Hamiltonian when B is small in the supplementary information.

Point-by-Point Response

Response to Reviewer # 1 (Remarks to the Author):

Comment 1.1 We would like to thank the authors for their exhaustive response to our comments. They have addressed all our comments by the revision of the manuscript and the supplementary material as well as by a detailed point-by-point response.

They clarified that experimental improvements compared to the previous work [13] included significantly longer coherence times, larger gate fidelities and improved control electronics. This was crucial for implementing deep quantum circuits required for realizing topological time crystals. They also clarified differences between the topologically ordered time crystals realized in this work and the Floquet symmetry-protected topological time crystals from their previous paper. These differences include the subharmonic response of global observables, robustness against all kinds of local perturbations and long-range entanglement. They motivated and explained the autocorrelation function of the logical Z operator. Its measurement in the experiment has also been clarified. The authors have convincingly answered our questions about the perimeter law and echo sequences by numerical simulations presented in the supplementary material.

We conclude that this work notably contributes to the investigation of the prethermal behavior of many-body systems. It presents impressive experimental results demonstrating the capabilities of the superconducting quantum processor. As all technical points have been convincingly addressed, we recommend the manuscript for publication in Nature Communications, provided that the authors address one remaining issue explained below that emerged in the revised manuscript.

Response 1.1 We thank the Reviewer for recommending publication, and for noting that our research “notably contributes to the investigation of the prethermal behavior of many-body systems” and “presents impressive experimental results demonstrating the capabilities of the superconducting quantum processor”. For the remaining one technical issue, we have carefully addressed it and modified the manuscript accordingly.

Comment 1.2 Technical comment: In the revised supplementary material, the authors clarify that the auto-correlation function $\langle X_L(0)X_L(t) \rangle$ of the logical X_L operator is not measured in the experiment. Instead, only the instantaneous value of X_L is measured. This is not problematic in itself, as they observe the subharmonic response of the autocorrelation function $\langle Z_L(0)Z_L(t) \rangle$ of the logical Z_L operator. However, in the caption of Fig. 2 and on page 15 of the supplementary material, the authors interpret the instantaneous value of X_L as its auto-correlation function. This interpretation is misleading and should be corrected. Moreover, on page 15 of the supplementary material, the authors claim that $\langle X_L(0)X_L(t) \rangle = 0$. This is not generally true. For vanishing on-site field $B = 0$, the logical X_L operator is a conserved quantity as it commutes with both H_1 and H_2 . As it squares to unity, $\langle X_L(0)X_L(t) \rangle = 1$ for any initial state in contrast to $\langle X_L(t) \rangle = 0$ for an initial computational basis state. This discrepancy indicates that interpreting the instantaneous value of X_L as its autocorrelation function is incorrect.

Response 1.2 We thank the Reviewer for raising this important point. We agree with the Reviewer that interpreting instantaneous values of X_L operators as its auto-correlation function is misleading since the initial states are polarized along the z -axis. We have modified the caption of Fig. 2 of the main text to clarify that Fig. 2a indeed shows the instantaneous expectation values for $\{X_{L_i}\}$ and the auto-correlators for $\{Z_{L_i}\}$. In addition, we explained the measurement details of the nonlocal X_L operators in Supplementary Section II.F, particularly for random bit-string initial states.

Reviewer # 2 (Remarks to the Author):

Comment 2.1 I co-reviewed this manuscript with one of the reviewers who provided the listed reports. This is part of the Nature Communications initiative to facilitate training in peer review and to provide appropriate recognition for Early Career Researchers who co-review manuscripts.

Response 2.1 We thank the Reviewer for reviewing our manuscript.

Response to Reviewer # 3 (Remarks to the Author):

Comment 3.1 I am satisfied with most of the responses of the authors to the previous round of reports. I did not realize that Fig. 4d already showed the expectation value of XL after one Floquet step, which addresses my concern about the relation of the Floquet eigenstate to the actual evolution as realized by the experiment.

Response 3.1 We thank the Reviewer for their time reviewing our revised manuscript and are happy to know that the Reviewer is “satisfied with most of the responses of the authors to the previous round of reports”.

Comment 3.2 However, I am still concerned about the interpretation of $S_{\text{topo}}(t)$. As I understand it, the authors’ response and addition to the main text appeals to the notion that $S_{\text{topo}}(t)$ being close to the surface code value of $\ln(2)$ means that the state is close to a surface code state (or one with the same topological order). Thus, the slow decay of $S_{\text{topo}}(t)$ could be interpreted as diagnosing the slow melting of topological order.

This interpretation of the topological entanglement entropy is an assumption, and should be stated as such. At the very least, the authors have skipped many steps in justifying this interpretation. If the state were guaranteed to be area law entangled and pure, then this interpretation would likely be valid. But a mixed and potentially volume law entangled state seems like it should be able to have any value of $S_{\text{topo}}(t)$ without necessarily having any relation to the corresponding topological order. $S_{\text{topo}}(t)$ in this paper goes through the value $-0.5 \cdot \ln(3)$, but the state is certainly not related to the $\nu=1/3$ Laughlin state.

That $S_{\text{topo}}(t)$ decays faster for $B=3.0$ than $B=0.1$ is a completely generic feature of making a stronger perturbation, and the same reasoning could have been applied to any observable which starts at 1 for $B=0$ and evolves to 0 after a quench to $B>0$. It does not imply that the numerical value of $S_{\text{topo}}(t)$ quantifies the extent to which the state remains topologically ordered.

If the authors insist that the topological entanglement entropy does quantify the remnant topological order in their state, then they must argue for this more carefully, and at least include their analysis in the supplemental material. Otherwise, it should be clearly stated that the relationship between the slow decay of $S_{\text{topo}}(t)$ and the existence of perturbatively dressed logical operators is an assumption which they do not intend to address in this work.

Figure 1: Experimentally measured fidelities of the four-qubit subsystem after up to five Floquet drive cycles t/T with small ($B = 0.1$) and large ($B = 3.0$) local perturbation strengths. The fidelities are obtained from the same datasets as Fig. 4c of the main text, where $F(\rho)(t \neq 0)$ is obtained by performing state tomography on a four-qubit subsystem and the average is over 12 random realizations; $F(\rho)(t = 0)$ is obtained via the same state tomography process and averaging over five repetitions of eigenstate preparation.

Response 3.2

The reviewer raises an excellent point here. We agree with the reviewer that the value of the topological entanglement entropy by itself is insufficient to fully determine whether the state is topologically ordered or not, and further information is required. We use topological entanglement entropy as an experimentally accessible indicator of the topological order, as is commonly done in the community [for example, see Science 374, 1237–1241 (2021), Commun. Phys. 7, 205 (2024)]. This is a reasonable assumption because the overlap between the topologically ordered initial state and the state obtained by evolving the initial state over several Floquet periods is large provided B is sufficiently weak (see Fig. 1). We elaborate on this point below.

- From the theoretical perspective of prethermalization, time evolving an eigenstate of the Floquet Hamiltonian at $B = 0$ with a modified Floquet drive at finite B should not completely destroy the topological order at early times provided B is small. In the limit of high-frequency drive, a Floquet system will enter a long-lived prethermal regime. In this case, the system's dynamics in stroboscopic time can be described by evolution under an effective static Hamiltonian H_{eff} , followed by a perfect spin-flip X ,

$$\begin{aligned}
 U_F &= X e^{-iH_{\text{eff}}T/2} + O(e^{-1/T}), \\
 H_{\text{eff}} &= - \sum_p \alpha_p A_p - \sum_q \beta_q B_q + \sum_k B_{k,x} \sigma_k^x + O(1/T).
 \end{aligned}$$

Note that X commutes with H_{eff} , allowing the evolution operator between even-numbered periods to be represented as $U(2nT) \approx e^{-iH_{\text{eff}}T}$, which is equivalent to a system evolved under the time-independent H_{eff} . With a small B , and hence small $B_{k,x}$, H_{eff} is still topologically ordered, so the evolution of an initially prepared $B = 0$ ground state $|E_+^{(0)}\rangle$ within the prethermal regime can be viewed at early times as a weak quench (from $B_{k,x} = 0$ to small but finite $B_{k,x}$) within the topologically ordered phase of H_{eff} . At later times but still within the prethermal window, higher-order corrections in B can destabilize the topological order. However, we expect that these corrections become significant on timescales $\gtrsim O(1/B^2)$, such that a finite time window remains in which the state is topologically ordered.

- In addition to the above theoretical interpretation, in our experiments, we can also measure state fidelities

$$F(\rho(t)) = \text{tr} \sqrt{\sqrt{\rho(t)}\rho^{\text{ideal}}\sqrt{\rho(t)}}, \quad (1)$$

for the four-qubit subsystem after up to five Floquet cycles (see Figure 1). At $B = 0.1$, the disorder-averaged $F(\rho)$ maintains a large value of ~ 0.88 after one period of Floquet evolution and decays slowly with more Floquet cycles, which indicates the topologically ordered initial state $|E_+^{(0)}\rangle$ tends to be maintained. In contrast, when B is large ($B = 3.0$), $F(\rho)$ drops to a floor value of ~ 0.5 immediately after one period of Floquet driving.

These additional analyses, together with the experimental results on topological entanglement entropy in the main text, provide convincing evidence to infer that the observed slow decay of $S_{\text{topo}}(t)$ at small B is a reasonable witness of the topological order in the state after Floquet evolution.

Following the suggestion of the reviewer, we have added the above analyses to the supplementary material. We have also explicitly added the following sentences in the main text: “We emphasize that the relationship between the observed nonzero topological entanglement entropy and the topological order in a general quantum state and away from equilibrium is complex. Here, we interpret the slow decay of S_{topo} as an indicator of slow melting of the topological order. This is supported by theoretical arguments based on prethermalization and by experimental measurements of the fidelity between the topologically ordered initial state and the state obtained by evolution over up to five Floquet periods, which remains large provided B is sufficiently weak (see Supplementary Section I.D).”